# Rescue of *Escherichia coli* auxotrophy by de novo small proteins

**Arianne M Babina[1], Serhiy Surkov[1†], Weihua Ye[1‡], Jon Jerlström-Hultqvist[1§], Mårten Larsson[1], Erik Holmqvist[2], Per Jemth[1], Dan I Andersson[1\*], Michael Knopp[1\*#]**

[1]Department of Medical Biochemistry and Microbiology, Uppsala University, Uppsala, Sweden; [2]Department of Cell and Molecular Biology, Uppsala University, Uppsala, Sweden

**\*For correspondence:**
Dan.Andersson@imbim.uu.se (DIA);
Knopp@embl.de (MK)

**Present address:** [†]CytaCoat, Stockholm, Sweden; [‡]Sprint Bioscience, Huddinge, Sweden; [§]Department of Cell and Molecular Biology, Uppsala University, Uppsala, Sweden; [#]European Molecular Biology Laboratory, Genome Biology Unit, Heidelberg, Germany

**Competing interest:** The authors declare that no competing interests exist.

**Abstract** Increasing numbers of small proteins with diverse physiological roles are being identified and characterized in both prokaryotic and eukaryotic systems, but the origins and evolution of these proteins remain unclear. Recent genomic sequence analyses in several organisms suggest that new functions encoded by small open reading frames (sORFs) may emerge de novo from noncoding sequences. However, experimental data demonstrating if and how randomly generated sORFs can confer beneficial effects to cells are limited. Here, we show that by upregulating *hisB* expression, de novo small proteins (≤50 amino acids in length) selected from random sequence libraries can rescue *Escherichia coli* cells that lack the conditionally essential SerB enzyme. The recovered small proteins are hydrophobic and confer their rescue effect by binding to the 5′ end regulatory region of the *his* operon mRNA, suggesting that protein binding promotes structural rearrangements of the RNA that allow increased *hisB* expression. This study adds RNA regulatory elements as another interacting partner for de novo proteins isolated from random sequence libraries and provides further experimental evidence that small proteins with selective benefits can originate from the expression of nonfunctional sequences.

## Editor's evaluation

This important study shows that small proteins encoded by randomized DNA sequences can be biologically active in bacteria, suggesting an evolutionary pathway for the creation of new biological functions. The authors use a combination of genetic and biochemical approaches to convincingly demonstrate a regulatory function for a randomly generated small protein in *Escherichia coli*. The work will be of interest to scientists working in the field of molecular evolution and cellular innovation.

## Introduction

Once overlooked, the study of small proteins is a rapidly growing field. Typically defined as polypeptides consisting of 50 or fewer amino acids, small proteins originate from the translation of distinct small open reading frames (sORFs), rather than from the cleavage of larger precursor proteins or synthesis via ribosome-independent mechanisms (*Hemm et al., 2020*; *Storz et al., 2014*). Recent advancements in genome and transcriptome sequencing, ribosome-profiling techniques, proteomics, and bioinformatic analyses have led to the discovery of numerous previously unannotated small proteins in all domains of life and efforts to elucidate the targets and functions of these proteins (*Andrews and Rothnagel, 2014*; *D'Lima et al., 2017*; *Hemm et al., 2008*; *Steinberg and Koch, 2021*; *Su et al., 2013*; *Weaver et al., 2019*; *Weidenbach et al., 2022*; *Yuan et al., 2018*). The

detection and characterization of small proteins in bacteria have been a particularly prolific research area over the past decade, and bacterial small proteins have been implicated in many fundamental physiological processes, including cell division, sporulation, lysis, transport, stress responses, virulence, antibiotic resistance, and cell-to-cell communication (for reviews, see *Duval and Cossart, 2017*; *Garai and Blanc-Potard, 2020*; *Hemm et al., 2020*; *Storz et al., 2014*). Nevertheless, despite the progress made in the identification and validation of an increasing number of small proteins with versatile cellular functions, much remains unknown about the origins, evolution, and phylogenetic distribution of these small genes and their encoded proteins.

In addition to the current underannotation of sORFs within genomic databases, the short lengths of the coding sequences and subsequent lack of conserved protein domains render it challenging to identify small protein orthologs and establish evolutionary relationships between small proteins across different organisms. For select bacterial small proteins encoded within operons containing larger, more conserved proteins, conservation of gene synteny and/or operon content has aided in the identification of orthologs in other bacteria (*Horler and Vanderpool, 2009*; *Storz et al., 2014*), but only a handful of bacterial small proteins have been found to traverse multiple phylogenetic classes. Instead, most appear to be poorly conserved and are limited to a single species or a few closely related bacteria (*Alix and Blanc-Potard, 2009*; *Storz et al., 2014*). This lack of conservation raises the question: where did these small protein-coding genes come from? Are they bona fide new genes that emerged independently or the remnants of genes that once encoded larger proteins?

A plausible mechanism for the de novo emergence of small protein-coding genes is the proto-gene model, wherein the transcription of noncoding DNA and subsequent ribosome association lead to the synthesis of novel proteins. Large-scale expression studies demonstrate pervasive transcription of non-genic stretches within characterized genomes (*Dinger et al., 2008*) and ribosome-profiling data indicate that many noncoding RNAs are engaged by the ribosome (*Carvunis et al., 2012*; *Wilson and Masel, 2011*), supporting the notion that the expression of randomly occurring sORFs from non-genic sequences can serve as a pool for the de novo selection of beneficial functions (*Baek et al., 2017*; *Hemm et al., 2008*; *Samayoa et al., 2011*). Furthermore, several reports show that genes recently emerged from noncoding DNA are often short in length, poorly conserved, composed primarily of hydrophobic amino acids, and tend to form alpha-helical domains – all of which are hallmark characteristics of most small proteins described to date (*Carvunis et al., 2012*; *Storz et al., 2014*).

Recent experimental work from our group has demonstrated that de novo small proteins with beneficial functions can be selected in vivo from completely random DNA sequence libraries. Due to their small size, the functions of the proteins recovered from these studies, as well as those of most naturally occurring small proteins, are mostly limited to interactions with preexisting cellular machineries or regulatory pathways rather than bona fide enzymatic activities. Specifically, our previously isolated de novo small proteins confer antibiotic resistance by altering cell permeability via direct interactions with the cell membrane (*Knopp et al., 2019*) or by activating a sensor kinase via protein–protein interactions (*Knopp et al., 2021*). Along similar lines, an earlier study by Digianantonio and Hecht using structurally constrained and partially randomized DNA libraries showed that selected proteins 102 amino acids in length can rescue an *Escherichia coli* auxotroph caused by the deletion of *serB* (*Digianantonio and Hecht, 2016*). While the precise mechanism of these semi-random proteins has not been elucidated, the dependence of the growth restoration on the deattenuation/increased transcription of the *his* operon and subsequent upregulation of the multi-copy suppressor, HisB, points toward a possible protein-RNA regulatory interaction as the underlying molecular basis of the rescue.

The abundance and mechanisms of RNA regulatory elements, especially those that regulate gene expression in response to direct protein binding, such as ribosomal protein leaders (*Fu et al., 2013*; *Zengel and Lindahl, 1994*) and Rho-dependent transcription terminators (*Banerjee et al., 2006*), render them promising potential targets for de novo small protein functionality. Additionally, the rescue of auxotrophies is a convenient means to probe for de novo small proteins with novel regulatory interactions as a number of biosynthetic operons are controlled by combinations of different regulatory proteins, RNA elements, and/or small molecules, and many auxotrophic phenotypes can often be suppressed by modulating the expression of alternate enzymes with moonlighting activities (*Patrick et al., 2007*).

To experimentally investigate the extent to which completely random, unconstrained, and/or noncoding sequences can serve as substrates for natural de novo gene evolution, we utilized random

sequence expression libraries (*Knopp et al., 2019*) to select for de novo small proteins that can restore the growth of an auxotrophic *E. coli* strain lacking the conditionally essential enzyme, SerB, and characterized the mechanisms responsible for the rescue phenotype. We isolated three small proteins from our screen that are less than or equal to 50 amino acids in length and are novel and distinct from those isolated from past studies. Our selected proteins confer their rescue effect by upregulating the alternative enzyme HisB, and the increase in *hisB* expression is likely caused by direct RNA-binding interactions with the regulatory 5′ end of the *his* operon mRNA transcript. In addition to their small size and gene regulatory roles, the recovered proteins exhibit other traits characteristic of most known naturally occurring small proteins, providing additional in vivo evidence that sORFs encoding novel beneficial functions can indeed originate de novo from previously noncoding and/or nonfunctional DNA sequences, without any preexisting structural or functional scaffolds. These findings add nucleic acids to the list of interacting partners for the novel de novo small proteins isolated from in vivo random sequence library screens.

## Results

### Selection of novel sORFs that restore growth of an auxotrophic *E. coli* mutant

We used a set of highly diverse expression vector libraries (*Knopp et al., 2019*) to screen for novel sORFs that confer rescue of the *serB* deletion mutant and other auxotrophic *E. coli* strains. The five libraries encode small proteins ranging from 10 to 50 amino acids in length (*Figure 1A*). Three of the libraries (rnd10, rnd20, rnd50a) encode repeats of NNB, where N encodes A, C, G, or T at equal ratios and B encodes only C, G, or T. This restriction was chosen to remove two of the three stop codons to increase the likelihood of obtaining full-length proteins within the sORFs, while maintaining amino acid ratios comparable to that of NNN repeats. Libraries 50b and 50c were biased to encode a higher fraction of primordial and hydrophilic residues in an effort to recapitulate potential coding sequences in early life (*Ring et al., 1972*) or generate proteins without a strong hydrophobic core that are structurally flexible (*Dunker et al., 2005*; *Dyson, 2016*), respectively. The inserts were expressed from an IPTG-inducible low-copy plasmid (an in-house construct denoted as pRD2 containing a p15A origin of replication). A strong ribosome-binding site, a start codon, and three stop codons (one per frame) ensured translational initiation and termination of the randomly generated inserts upon IPTG induction. The total diversity of the libraries was approximately $5.8 \times 10^8$.

We used these libraries to select for plasmid clones that could restore growth of the Δ*serB* auxotroph and other auxotrophic *E. coli* mutants on minimal medium. The strains used were single-gene knockout mutants from the KEIO collection (*Baba et al., 2006*). A previous study showed that 155 KEIO strains are unable to grow on M9 minimal medium (*Baba et al., 2006*; *Joyce et al., 2006*; *Patrick et al., 2007*); we selected 74 of these strains for our screen (*Supplementary file 1a*). These mutants included (i) strict or 'tight' auxotrophs that do not form colonies or residual growth on plates even after 2 wk of incubation, (ii) 'leaky' auxotrophs that are able to grow after an extended incubation time, and (iii) auxotrophs with high reversion frequencies, which are generally not able to grow even after prolonged incubation periods, but have a higher frequency of reversion via chromosomal mutations. We transformed all five expression libraries as well as empty vector controls into these 74 auxotrophic mutants and selected three variants that enabled growth on M9 minimal medium (*Figure 1B*). While multi-copy suppression of these auxotrophies has previously been shown to be common (*Patrick et al., 2007*), we were only able to isolate randomly generated sORFs that could rescue the Δ*serB* auxotroph from our screens. The inserts encoded in the three selected variants cause little to no reduction in fitness when expressed during exponential phase growth in rich medium (*Figure 1—figure supplement 1*) and were designated <u>h</u>is <u>d</u>eattenuating <u>protein</u>(s) 1–3 (Hdp1-Hdp3).

### Small proteins that confer rescue are alpha-helical and share a putative sequence motif

To determine whether the rescue of the *serB* deletion mutant is facilitated by expression of the mRNA or the encoded protein, we constructed Hdp1 and Hdp2 variants containing frameshifts and premature stop codons. None of these constructs were able to restore growth of the auxotrophic mutant on minimal medium, indicating that the translated proteins are responsible for the rescue

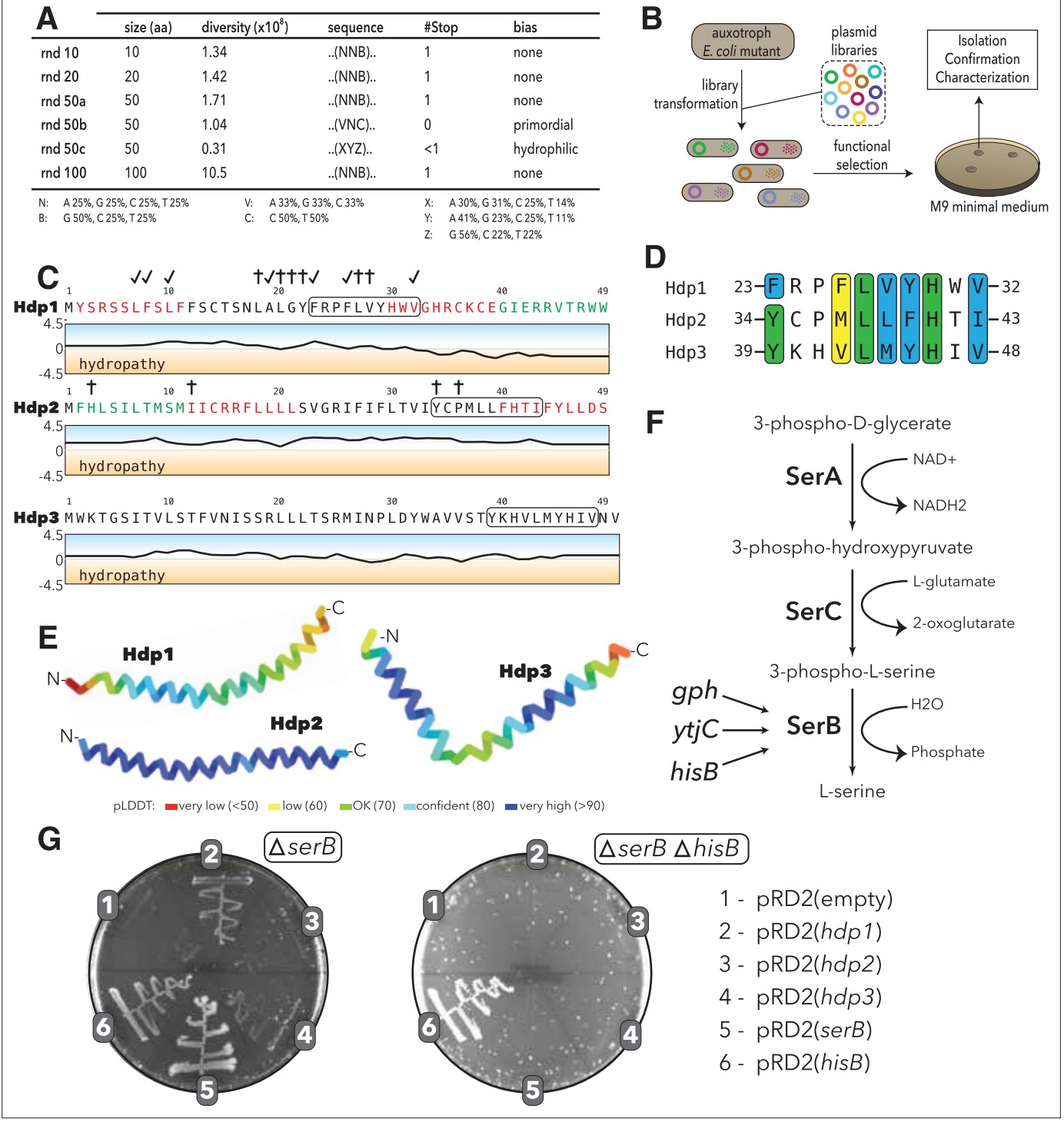

**Figure 1.** Experimental setup and sequence characteristics of the isolated small proteins. (**A**) Libraries cloned into the expression vector pRD2. (**B**) Plasmid transformation into auxotrophic mutants and selection for rescue of auxotrophic mutants. (**C**) Hydropathy profiles of the three isolated small proteins (Hdp1–3). Colored amino acids denote residues that could (green) or could not (red) be removed while maintaining functionality. Loss-of-function mutations for Hdp1 and Hdp2 are indicated by daggers (†), and mutations that retain rescue activity are indicated by check marks; mutations are further described in **Supplementary file 1b**. The box spanning 10 amino acids denotes a potential region of similarity between the small proteins. (**D**) Sequence alignment showing the putative region of similarity shared between the three small proteins. Green coloring indicates identical amino acids, while blue and yellow coloring indicate strongly and weakly similar amino acids, respectively. (**E**) Structure prediction of Hdp1–3 obtained from AlphaFold (**Jumper et al., 2021**; **Mirdita et al., 2022**). Colors represent the per-residue prediction confidence level (pLDDT), based on the lDDT-cα metric (**Mariani et al., 2013**). (**F**) The three enzymes SerA, SerC, and SerB catalyze the last step in L-serine biosynthesis. Essentiality of SerB on minimal

*Figure 1 continued on next page*

*Figure 1 continued*

medium can be suppressed by overexpression of the phosphatases Gph, YtjC, or HisB. (**G**) Growth of *E. coli* Δ*serB* and Δ*serB* Δ*hisB* deletion strains carrying either the empty pRD2 plasmid or pRD2 encoding Hdp1-3, SerB, or HisB on M9 minimal medium supplemented with 50 μg/ml ampicillin, 0.2% glucose, and 1 mM IPTG. Pictures taken after 1-wk incubation at 37°C. Growth on M9-glucose indicates the ability of the cloned construct to rescue Δ*serB* auxotrophy.

The online version of this article includes the following source data and figure supplement(s) for figure 1:

**Source data 1.** Random sequence libraries cloned into the expression vector pRD2.

**Figure supplement 1.** Growth rates of strains expressing Hdps relative to empty pRD2 control.

**Figure supplement 2.** Characterization of Hdp1 mutations and truncations.

**Figure supplement 3.** Characterization of Hdp2 mutations and truncations.

**Figure supplement 4.** Hdp structure predictions using JPred4.

**Figure supplement 5.** Hdp complementation of various deletion strains.

(*Figure 1—figure supplements 2 and 3*). Furthermore, we generated variants encoding the proteins using alternative codons, which extensively changes the nucleotide sequence while maintaining the amino acid sequence. Recoded Hdp2 retained function, demonstrating that the rescue is indeed mediated by the translated sORF (*Figure 1—figure supplement 3*). However, recoding Hdp1 resulted in a loss of function, likely due to effects on expression levels and mRNA stability. Therefore, to further demonstrate that the functionality of this insert is also dependent on the encoded protein rather than the mRNA transcript, we constructed an additional Hdp1 variant in which the start codon was removed. This variant did not allow growth on minimal medium, confirming that protein expression is essential for rescue (*Figure 1—figure supplement 2*).

The encoded small proteins are hydrophobic 49- and 50-mers (*Figure 1C*) and truncation experiments showed that 10 amino acids could be removed from the C- and N-terminus of Hdp1 and Hdp2, respectively, with maintained functionality (*Figure 1—figure supplements 2 and 3*). Multiple sequence alignments using Clustal Omega (*Chojnacki et al., 2017*) revealed a putative motif of 10 amino acids that was shared among all three proteins (*Figure 1D*). This possible motif has not been described previously and is not recognized in the Pfam database of protein families (*El-Gebali et al., 2019*). All three isolated proteins are predicted to be mainly alpha-helical with high confidence by AlphaFold as well as JPred4 (*Drozdetskiy et al., 2015*; *Jumper et al., 2021*; *Mirdita et al., 2022*; *Figure 1E*, *Figure 1—figure supplement 4*). To examine the potential functional role of this region of similarity and define other functional regions of the proteins, we performed random mutagenesis of the *hdp1* and *hdp2* genes and screened for mutant variants that were unable to rescue the Δ*serB* mutant on minimal medium. Some loss-of-function mutations observed were clustered in or near the similarity region, suggesting that it may have a role in rescue (*Figure 1C*, *Figure 1—figure supplements 2 and 3*; *Supplementary file 1b*). Nevertheless, further studies are needed to verify and characterize the role of this putative motif in the Hdp-mediated rescue of a Δ*serB* mutant on minimal medium.

## Rescue of the Δ*serB* auxotrophy in *E. coli* K12 is *hisB*-dependent

SerB is a phosphatase that catalyzes the conversion of 3-phospho-L-serine to L-serine in the final step of L-serine biosynthesis (*Ravnikar and Somerville, 1987*). To test whether the isolated proteins can bypass the normal pathway for L-serine biosynthesis, we examined whether they could rescue auxotrophies caused by the deletion of the enzymes upstream in this linear pathway (*Figure 1F*). On minimal medium, none of the Hdp proteins enabled growth of either a Δ*serA* or Δ*serC* mutant, which encode a dehydrogenase and an aminotransferase, respectively, indicating that the Hdps do not re-route the metabolism to synthesize serine via a different pathway, but rather they relieve the need for SerB (*Figure 1—figure supplement 5*).

*E. coli* encodes 23 cytoplasmic haloacid dehydrogenase (HAD)-like hydrolases (including SerB), which share limited sequence similarity but have strongly overlapping substrate specificities (*Kuznetsova et al., 2006*). To determine whether expression of the selected small proteins could functionally replace any of the other HAD-like phosphatases, we tested the growth defects of individual HAD-knockout mutants. Besides Δ*serB*, only Δ*hisB* exhibited an auxotrophic phenotype.

However, only the ΔserB auxotrophic mutant could be rescued by expression of the isolated proteins (*Figure 1G*).

Past studies showed that the overexpression of two HAD-like phosphatases, HisB and Gph, as well as the nonrelated putative phosphatase YtjC, can rescue ΔserB auxotrophy (*Patrick et al., 2007*; *Yip and Matsumura, 2013*). We therefore tested whether the small protein-mediated rescue is dependent on the presence of any of these three proteins. While removal of YtjC and Gph did not affect the rescue, deletion of *hisB* abolished the ability of Hdp1-Hdp3 to rescue the ΔserB auxotrophy (*Figure 1G*, *Figure 1—figure supplement 5*). Based on these findings, we hypothesized that the Hdps rescue the lack of SerB by upregulating expression of HisB, which can functionally replace SerB and thereby restore L-serine biosynthesis and growth on minimal medium (*Figure 1F*). This is in agreement with the previous Digianantonio and Hecht study, which showed that sequence libraries encoding semi-random proteins that fold into predefined four-helix bundles could be used to select proteins that upregulate HisB to rescue SerB deficiency (*Digianantonio and Hecht, 2016*; *Fisher et al., 2011*).

## The *his* operator region is required for the Hdp-mediated rescue of ΔserB auxotrophy

Expression of the *his* operon, consisting of the structural genes *hisG*, *hisD*, *hisC*, *hisB*, *hisH*, *hisA*, *hisF*, and *hisI*, is regulated by a transcriptional attenuator located near the 5′ end of the *his* operon mRNA transcript that responds to levels of charged tRNA$^{His}$ (*Figure 2A*). Under histidine-rich conditions, a histidine-rich leader peptide (HisL) is synthesized and a terminator hairpin forms, resulting in transcription termination. However, under histidine-poor conditions, the ribosome will stall at the HisL histidine codons due to the lack of charged tRNA$^{His}$. As a result, an anti-terminator hairpin forms that allows continued transcription and subsequent expression of the downstream structural genes (*Artz and Broach, 1975*; *Blasi and Bruni, 1981*; *Johnston et al., 1980*; *Kasai, 1974*). Additionally, the operon promoter is activated upon induction of the stringent response, offering various potential targets with which the Hdps could interact. For simplicity, we designated the aforementioned regulatory region of the *his* operon as the '*his* operator (*his*$_{operator}$),' comprised of the *hisL* leader peptide gene and the transcription attenuator and under the control of the P$_{hisL}$ promoter (*Figure 2A*).

To test whether increased expression of HisB by the Hdps is dependent on the promoters and/or regulatory region of the *his* operon transcript, we constructed various transcriptional fusions of the *his* operator to the native *lacZ* gene on the chromosome of the ΔserB strain to monitor activity in vivo. Hdp1 and Hdp3 expression caused an almost 20-fold increase in β-galactosidase activity of the reporter strain carrying the full-length P$_{hisL}$-*his*$_{operator}$-*lacZ* fusion (under the control of the native P$_{hisL}$ promoter) (*Figure 2B*). Only a twofold increase was observed for Hdp2, consistent with our observations of ΔserB strain growth on minimal medium where cells expressing Hdp1 and Hdp3 grew faster than those expressing Hdp2 (*Figure 1G*). Correspondingly, a nonfunctional Hdp1 variant containing a L27Q substitution within the similarity region that was derived from the random mutagenesis screens (i.e., did not rescue the ΔserB mutant on minimal medium) did not increase the β-galactosidase activity of the P$_{hisL}$-*his*$_{operator}$-*lacZ* fusion reporter strain (*Figure 2B*, *Figure 1—figure supplement 2*; *Supplementary file 1b*). To examine the specificity of the Hdp mechanism of action, we also assayed the β-galactosidase activity of a reporter strain carrying the full-length *thr* operator sequence transcriptionally fused to the *lacZ* gene on the chromosome. In *E. coli*, the *thr* biosynthetic operon is regulated by an operator region that is similar in length, structure, and mechanism to that of the *his* operon (*Kolter and Yanofsky, 1982*). As expected, Hdp1 expression had little impact on the β-galactosidase activity of the P$_{thrL}$-*thr*$_{operator}$-*lacZ* reporter strain (*Figure 2B*). Overall, these data confirm that the Hdps rescue *serB* auxotrophy by increasing *hisB* expression through altering the regulation of the *his* operon via the *his* operator region. Due to its robust activity in vivo and the clear negative effect of the L27Q substitution, Hdp1 was primarily used for all subsequent characterization studies.

We next examined whether the stringent response is involved in the mechanism of the Hdps as this pathway is a known regulator of many biosynthetic operons, including the *his* and *thr* operons (*Paul et al., 2005*; *Riggs et al., 1986*; *Stephens et al., 1975*). In addition to the negligible impact Hdp1 expression had on the β-galactosidase activity of the full-length P$_{thrL}$-*thr*$_{operator}$-*lacZ* reporter strain, Hdp1 expression did not affect the β-galactosidase activity of a *lacZ* transcriptional fusion under the control of the *rrnB* P1 promoter and accompanying regulatory sequence, which is also regulated by

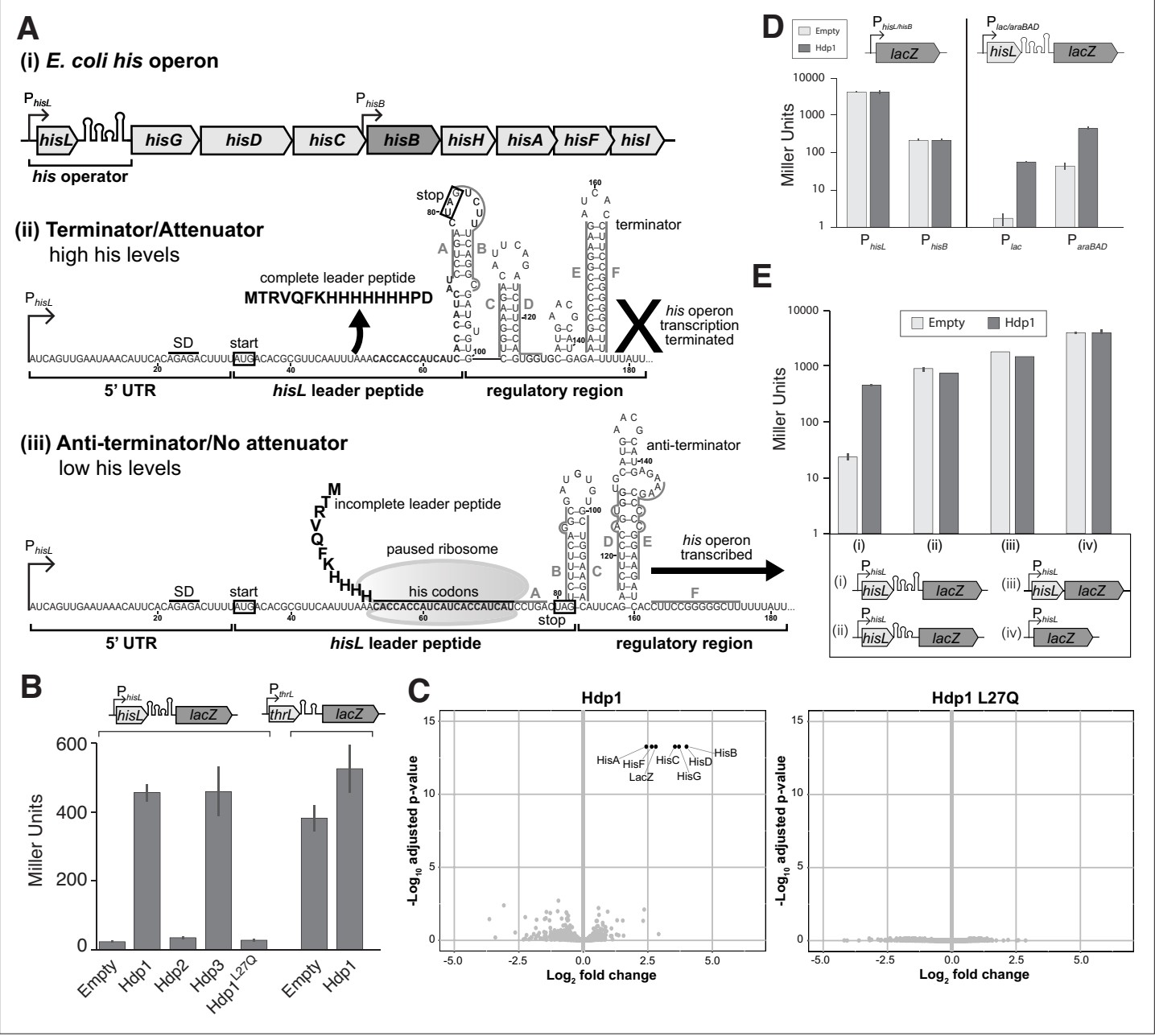

**Figure 2.** Overview of *his* operon regulation in *E. coli* and regulatory activity of the Hdps. (**A**) (i) The regulatory region and structural genes of the *his* operon, including *hisB* (highlighted in dark gray), (ii) the *his* operator and RNA secondary structure under histidine-rich conditions, and (iii) the *his* operator and RNA secondary structure under histidine starvation. P*hisL* is the promoter; *hisL* is the leader peptide; SD is the Shine-Dalgarno sequence for *hisL*; the start and stop codons of the *hisL* coding region are boxed. Nucleotides are numbered from the transcription start site, +1. RNA secondary structures are adapted from ***Johnston et al., 1980***; ***Kolter and Yanofsky, 1982***. (**B**) β-Galactosidase activity (in Miller Units) of the strain carrying the full-length P*hisL*-*his*operator-*lacZ* reporter upon expression of Hdp1-3 and the Hdp1 L27Q mutant, and the strain containing a P*thrL*-*thr*operator-*lacZ* reporter upon the expression of Hdp1. (**C**) 'Volcano plots' showing significant changes in protein abundance in the Δ*serB* mutant containing the full-length P*hisL*-*his*operator-*lacZ* reporter upon expression of Hdp1 or the Hdp1 L27Q nonfunctional mutant versus the empty plasmid control. An adjusted p-value cut-off of 0.01 and log$_2$ fold change cut-off of 2 was used. Due to their small size, HisL and the Hdp variants were not recovered in these proteomics samples. Total proteome analysis was performed with at least three biological replicates for each strain. Some data points overlap. (**D**) β-Galactosidase activity of various *lacZ* reporter constructs under the control of different promoters upon expression of Hdp1 versus the empty plasmid control: *lacZ* transcriptional fusions under the control of the native *hisL* and *hisB* promoters (no *his* operator regulatory sequence) and the full-length *his*operator-*lacZ* reporter construct under the control of the IPTG-inducible P*lac* or arabinose-inducible P*araBAD* promoters. (**E**) β-Galactosidase activity of various truncated P*hisL*-*his*operator-*lacZ* reporter constructs upon expression of Hdp1 versus the empty plasmid control. Refer to ***Supplementary file 1c*** for the nucleotide sequences

*Figure 2 continued on next page*

*Figure 2 continued*

of each *lacZ* reporter construct. For the β-galactosidase data presented in panels (**B, D, E**), the values reported represent the mean of three or more independent biological replicates; error bars represent the standard deviation.

The online version of this article includes the following figure supplement(s) for figure 2:

**Figure supplement 1.** Hdp regulatory activity is independent of the stringent response.

**Figure supplement 2.** Protein abundance of strains expressing Hdps relative to the empty plasmid control strain.

the stringent response (*Aseev et al., 2014*; *Paul et al., 2004*; *Paul et al., 2005*; *Figure 2—figure supplement 1A*). Moreover, the rescue effect of Hdp1 was not affected by the deletion of the stringent response genes *relA* and *dksA* in the P$_{hisL}$-*his*$_{operator}$-*lacZ* fusion reporter strain (*Paul et al., 2004*; *Turnbull et al., 2019*; *Figure 2—figure supplement 1B*).

To further evaluate whether Hdp expression activates cellular stress response mechanisms, we also performed global proteome analyses of cells expressing Hdp1-3 or the Hdp1 L27Q nonfunctional mutant versus those with an empty plasmid control. We used the Δ*serB* strain carrying the full-length P$_{hisL}$-*his*$_{operator}$-*lacZ* fusion as the test strain (DA57390, *Supplementary file 1c*). The abundances of only seven proteins were significantly increased upon expression of Hdp1 and Hdp3, all of which were under the regulation of the *his* operator (i.e., the native *his* operon and modified *lac* operon) (*Figure 2C*, *Figure 2—figure supplement 2*). However, we did not detect a significant increase in His or modified Lac protein abundance upon expression of Hdp2. Since Hdp2 causes the weakest growth restoration in a Δ*serB* strain, the effect on HisB abundance is expected to be low. It is possible, albeit unlikely, that Hdp2 rescues via a different mechanism. Consistent with our β-galactosidase activity data, no significant changes to protein abundance were observed upon expression of the Hdp1 L27Q mutant. Taken together, these data suggest that the Hdps do not induce *his* operon expression through the activation or upregulation of stress response pathways and that Hdps have a high specificity in their action and do not cause global alterations in gene expression.

To assess whether the Hdps alter HisB expression through direct interactions with either of the two native *his* operon promoters, we generated *lacZ* transcriptional fusions (with no *his* operator regulatory sequence) under the control of the P$_{hisL}$ promoter, which is the primary *his* operon promoter located upstream of the *hisL* leader peptide gene, and the P$_{hisB}$ promoter, an internal promoter for the operon located just upstream of the *hisB* gene (*Grisolia et al., 1983*). No increase in β-galactosidase activity was observed upon Hdp1 expression for either reporter strain carrying the P$_{hisL}$-*lacZ* or P$_{hisB}$-*lacZ* transcriptional fusions (*Figure 2D*). However, Hdp1 was still able to increase the β-galactosidase activity of *his*$_{operator}$-*lacZ* transcriptional fusions in which the native P$_{hisL}$ promoter was replaced with either the P$_{lac}$ promoter or the P$_{araBAD}$ promoter (induced upon the addition of IPTG or arabinose, respectively), showing that the Hdp rescue mechanism does not require the native *his* operon P$_{hisL}$ and P$_{hisB}$ promoters.

Finally, no increase in β-galactosidase activity was observed upon Hdp1 expression in strains carrying truncated versions of the *his* operator fused to *lacZ* (*Figure 2E*). An Hdp1-mediated increase in β-galactosidase activity was also not observed for a full-length P$_{hisL}$-*his*$_{operator}$-*lacZ* reporter strain containing a missense mutation within the *hisL* leader peptide coding sequence (Q5Stop) (*Figure 3—figure supplement 1C*). Taken together, our data demonstrate that the Hdps require the complete and fully functional *his* operator regulatory region, consisting of the full-length and intact *hisL* gene and transcription attenuator sequence, to exert their rescue effect.

## Hdp1 binds the *his* operator mRNA

Based on our β-galactosidase activity data, we hypothesized that the Hdps likely modulate *his* operon expression through direct interaction with the 5′ end operator region of the *his* operon mRNA transcript. The original Hdp1 protein exhibited very low solubility in water (<0.2 mg/ml), initially precluding us from performing certain binding assays. To circumvent this problem, a functional and more hydrophilic Hdp1 variant with increased solubility in water (>1 mg/ml), named Hdp1-optimized (Hdp1$_{opt}$), was derived from the mutagenesis experiments. We also generated an Hdp1$_{opt}$ variant containing the previously described L27Q substitution, which completely abolished Hdp1 activity in vivo, to serve as a control for our binding experiments (*Figure 3—figure supplement 1*; *Supplementary file 1b*).

To demonstrate interactions between Hdp1 and the *his* operator RNA in vivo, we performed co-immunoprecipitation (co-IP) assays with cell lysates from our Δ*serB* reporter strain carrying the full-length P*hisL*-*his*operator-*lacZ* fusion and expressing Hdp1opt and Hdp1opt L27Q variants with C-terminal HA tags (*Figure 3—figure supplements 1 and 2*). Co-IP experiments were also performed using an untagged Hdp1opt variant as a background/normalization control and interacting RNA transcripts of interest were detected and quantified via RT-qPCR. Overall, we observed a marked enrichment of select RNA transcripts whose expression is directly regulated by the 5′ *his* operator region (i.e., genes within the native *his* operon and *his*operator-*lacZ* reporter operon) in the HA-tagged Hdp1opt pull-down fractions (*Figure 3A*). Specifically, *hisL*, whose coding region is contained within the *his* operator regulatory region, exhibited fivefold enrichment in the pull-down samples, and *his* operon genes *hisG* (the first protein-coding gene immediately downstream from the *his* operator region) and *hisB* demonstrated approximately 20- and 14-fold enrichment, respectively. Similarly, the *his*operator-*lacZ* reporter transcript showed eightfold enrichment in the Hdp1opt pull-down fractions.

In contrast, less enrichment was observed for the control *thrL* and *thrA* RNA transcripts (2- and 3.5-fold, respectively), whose protein-coding regions are either within or immediately downstream from the *thr* operator region, which was used as a specificity control for the β-galactosidase activity assays. Co-IP experiments performed with the tagged, nonfunctional Hdp1opt L27Q mutant variant also demonstrated less enrichment for all target *his* and *lacZ* RNA transcripts of interest. Although these data demonstrate possible weak in vivo interactions between Hdp1opt and the *thr*operator RNA and between Hdp1opt L27Q and the *his* and *lacZ* transcripts, the lower fold-enrichment for both data sets suggests that these potential interactions are negligible enough to impede any measurable regulatory activity in vivo and the subsequent rescue of Δ*serB* strain growth on minimal medium.

It is interesting to note that despite the fact that *hisL* is contained within the *his* operator region (the primary target for Hdp1opt binding) and there are two copies of *hisL* present in the reporter strain used for the experiments, the fold-enrichment for *hisL* is slightly lower than that of the other downstream transcripts regulated by the *his* operator in the pull-downs performed with both the functional and nonfunctional versions of Hdp1opt. This could potentially be due to differences in transcript stability and/or reverse transcription efficiency as *hisL* is a very small gene located next to a highly structured region on the 5′-most ends of two very long transcripts. This is also likely the reason why the same trend is apparent in the pull-down data for the *thrL* versus the *thrA* transcripts as *thrL* has a genomic context similar to that of *hisL*. Nevertheless, the overall trends observed in the co-IP data are in agreement with those from our proteomics and β-galactosidase activity experiments and indicate that Hdp1 directly interacts with RNA transcripts regulated by the 5′ *his* operator region in vivo.

Binding of Hdp1opt to the full-length *his* operator RNA was further confirmed and characterized in vitro by electrophoretic mobility shift assays (EMSAs). The $K_d$ value from a 1:1 binding model was estimated as $0.63 \pm 0.13$ µM. Consistent with our above in vivo data, EMSAs performed with the nonfunctional Hdp1opt L27Q variant demonstrated a fivefold reduction in binding affinity to the *his* operator RNA ($K_d = 3.3 \pm 0.71$ µM), and assays performed with the 'wild-type' Hdp1opt and the nonspecific control *thr* operator RNA exhibited an almost threefold reduction in binding affinity in vitro ($K_d = 1.7 \pm 0.37$ µM) (*Figure 3B and C*). However, while a maximum bound fraction of 1 is expected from the experimental setup, the fitted maximum fraction bound was $1.45 \pm 0.13$ for the L27Q mutant protein. This value results from a slightly sigmoidal shape of the binding curve, suggesting positive cooperativity. We therefore fitted an equation accounting for positive cooperativity to all three data sets, resulting in $K_d$ values of $0.54 \pm 0.08$ µM (Hdp1opt with the *his* operator RNA), $1.8 \pm 0.26$ µM (Hdp1opt L27Q with the *his* operator RNA), and $1.28 \pm 0.27$ (Hdp1opt with the *thr* operator control RNA), with Hill coefficients of around 1.4 (*Figure 3—figure supplement 3*; *Supplementary file 1d and e*). While the number of data points collected precludes any strong conclusion regarding the binding model, the EMSAs support our in vivo data by demonstrating (i) a direct binding interaction between the Hdp1opt protein and the *his* operator RNA, (ii) Hdp1opt displays a higher affinity for the *his* RNA than the *thr* RNA, and (iii) the 'wild-type' Hdp1opt has a greater affinity for the *his* RNA than the L27Q mutant protein.

## Hdp1 binding alters *his* operator RNA secondary structure

To assess whether portions of the full-length *his* operator RNA undergo conformational changes upon Hdp1opt binding and/or which regions are necessary for Hdp1opt interaction, we performed nuclease

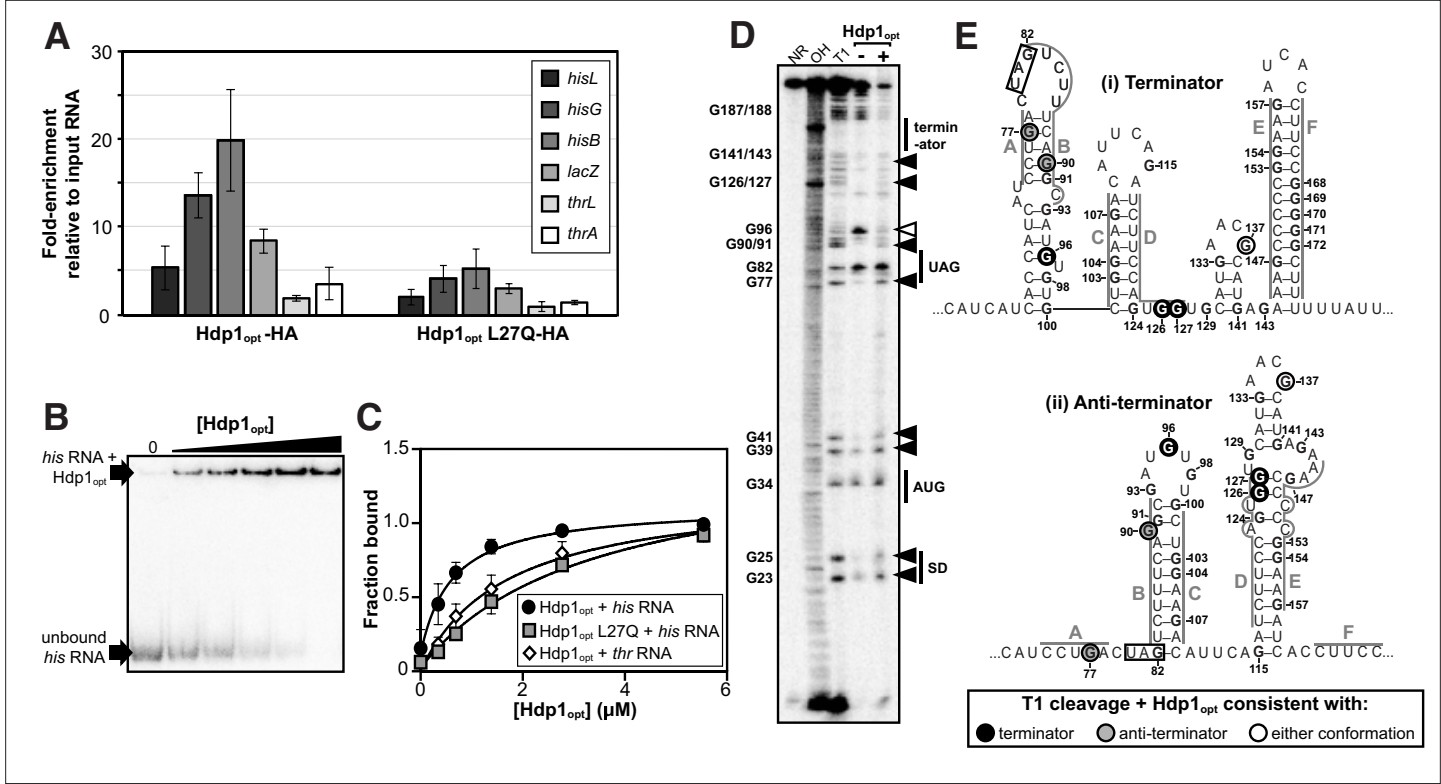

**Figure 3.** Characterization of Hdp1opt binding interactions with the *his* operator mRNA. (**A**) Enrichment of select RNA transcripts in the HA-tagged Hdp1opt and Hdp1opt L27Q mutant pull-down samples, as quantified by RT-qPCR. *thrL* and *thrA* are nonspecific control RNAs. RT-qPCR data from pull-downs performed with untagged Hdp1opt was used as a normalization/background control. The data reported is the mean of three independent experimental replicates; error bars represent the standard deviation. (**B**) Representative electrophoretic mobility shift assay (EMSA) of the full-length *his* operator RNA in the presence of increasing concentrations of Hdp1opt (0–5.5 μM). Refer to ***Figure 3—figure supplement 3*** for representative EMSA gels for all RNA and protein combinations assayed. All EMSAs were repeated independently three or more times. (**C**) Binding curves obtained from fitting the data quantified from the EMSAs to a standard 1:1 hyperbolic binding model. Each data point represents the mean of three or more independent experimental replicates; error bars represent the standard error. Some error bars fall within the boundaries of the markers. For the parameters used and the EMSA data fit to a binding model accounting for positive cooperativity, refer to ***Supplementary file 1d and e***. (**D**) RNase T1 probing gel for the full-length *his* operator RNA in the absence and presence of Hdp1opt (0 and 5.5 μM, respectively). NR denotes RNA subject to no reaction, OH indicates partial alkaline hydrolysis, and T1 is an RNase T1 digest of the RNA under denaturing conditions used to map the RNA sequence. Numbering of G nucleotides is shown on the left. Arrows highlight changes in RNA cleavage in the presence of Hdp1opt: black arrows indicate nucleotides with increased cleavage and white arrows indicate reduced cleavage. Sequence and/or structure characteristics of the *his* operator RNA are also indicated on the right (i.e., the Shine-Dalgarno sequence [SD], start codon [AUG], and stop codon [UAG] of the *hisL* leader peptide coding sequence). This experiment was repeated three independent times with similar results; a representative gel is shown. The data from additional replicates can be found in ***Figure 3—figure supplement 4***. (**E**) Nucleotides demonstrating a change in RNase T1 cleavage in the presence of Hdp1opt mapped to both the (i) terminator and (ii) anti-terminator conformations of the *his* operator RNA. The conformation consistent with cleavage in the presence of Hdp1opt is color-coded according to the legend; the *hisL* stop codon is boxed. Nucleotides are numbered from the transcription start site, +1.

The online version of this article includes the following source data and figure supplement(s) for figure 3:

**Source data 1.** Source data for ***Figure 3B and D***.

**Figure supplement 1.** Characterization of Hdp1opt, the Hdp1 variant optimized for increased water solubility.

**Figure supplement 2.** Western blot detection of HA-tagged Hdp1 variants following co-immunoprecipitation assays.

**Figure supplement 2—source data 1.** Western blot containing protein fractions from the different steps of the co-immunoprecipitation experiments.

**Figure supplement 2—source data 2.** Western blot containing protein fractions from the different steps of the co-immunoprecipitation experiments.

**Figure supplement 2—source data 3.** Western blot containing protein fractions from the different steps of the co-immunoprecipitation experiments.

**Figure supplement 2—source data 4.** Western blot containing protein fractions from the different steps of the co-immunoprecipitation experiments.

**Figure supplement 2—source data 5.** Western blot containing protein fractions from the different steps of the co-immunoprecipitation experiments.

**Figure supplement 2—source data 6.** Western blot containing protein fractions from the different steps of the co-immunoprecipitation experiments.

**Figure supplement 3.** Electrophoretic mobility shift assay (EMSA) gels and binding curves from fitting of different binding models to the EMSA data.

*Figure 3 continued on next page*

*Figure 3 continued*

**Figure supplement 3—source data 1.** Electrophoretic mobility shift assay (EMSA) of the full-length *his* operator RNA in the presence of increasing concentrations of Hdp1$_{opt}$.

**Figure supplement 3—source data 2.** Electrophoretic mobility shift assay (EMSA) of the full-length *his* operator RNA in the presence of increasing concentrations of Hdp1$_{opt}$.

**Figure supplement 3—source data 3.** Electrophoretic mobility shift assay (EMSA) of the full-length *his* operator RNA in the presence of increasing concentrations of Hdp1$_{opt}$ (left) or the Hdp1$_{opt}$ L27Q mutant (right).

**Figure supplement 3—source data 4.** Electrophoretic mobility shift assay (EMSA) of the full-length *his* operator RNA in the presence of increasing concentrations of the Hdp1$_{opt}$ L27Q mutant.

**Figure supplement 3—source data 5.** Electrophoretic mobility shift assay (EMSA) of the full-length *thr* operator RNA in the presence of increasing concentrations of Hdp1$_{opt}$.

**Figure supplement 3—source data 6.** Electrophoretic mobility shift assay (EMSA) of the full-length *thr* operator RNA in the presence of increasing concentrations of Hdp1$_{opt}$.

**Figure supplement 4.** RNase T1 probing experimental replicates.

**Figure supplement 4—source data 1.** Uncropped RNase T1 probing gel for the full-length *his* operator RNA in the absence and presence of Hdp1$_{opt}$ (0, 0.69, and 5.5 µM).

protection assays with RNase T1 in the presence and absence of Hdp1$_{opt}$ (±5.5 µM Hdp1$_{opt}$, the maximum concentration used in the EMSAs). RNase T1 specifically cleaves RNA at single-stranded or unprotected G nucleotides. The *his* operator RNA can adopt two mutually exclusive conformations (*Johnston et al., 1980*; *Kolter and Yanofsky, 1982*), and probing the RNA structure in the absence of Hdp1$_{opt}$ showed hallmarks of both conformations (*Figure 3D and E*, *Figure 3—figure supplement 4*). For instance, protection of cleavage at G77 and G90 is consistent with the terminator conformation, while the strong cleavage at G96 and protection at G126-127 is consistent with the anti-terminator structure, suggesting a mixture of the two conformations, or possibly alternative conformations, in the RNA sample.

Several regions of the *his* operator RNA demonstrated increased cleavage in the presence of Hdp1$_{opt}$, and the cleavage pattern suggests a potential transition between the terminator and the anti-terminator conformations. Specifically, while the increased cleavage of nucleotides G126 and G127 in the presence of Hdp1$_{opt}$ is consistent with the terminator conformation of the *his* operator RNA, the increased cleavage of nucleotides G77 and G90 is consistent with the anti-terminator structure (*Figure 3D and E*). Interestingly, only one nucleotide, G96, exhibited reduced cleavage upon the addition of Hdp1$_{opt}$. The protection of G96 in the presence of the protein is consistent with the terminator conformation of the *his* operator RNA; however, it may also indicate a region of the RNA that is directly shielded by protein binding.

The intensity of T1-cleavage products corresponding to the *hisL* Shine-Dalgarno sequence and start codon (nucleotides G23-G41) also increased in the Hdp1$_{opt}$-bound RNA. In addition to the impact the Hdps have on *his* operon transcription, it is also possible that the 5′ region of the RNA becomes more accessible to the ribosome in the presence of Hdp1$_{opt}$, thus facilitating more efficient translation of the *his* operon and further increasing HisB expression. However, the biological relevance of this T1-cleavage pattern is difficult to discern, as the 5′ region of the *his* operator RNA is predicted to remain relatively unstructured in either conformation (*Johnston et al., 1980*; *Kolter and Yanofsky, 1982*).

Some highly structured regions could not be fully resolved in either the T1 probing assays or the denaturing T1 reference ladder, particularly the nucleotides that comprise the terminator stem (E+F, nucleotides G153-G171) (*Figure 3D and E*). This indicates that the G-C-rich stem is difficult to disrupt, even under denaturing conditions and with a high RNase T1 concentration (*Johnston et al., 1980*; *Kolter and Yanofsky, 1982*). Similarly, the region corresponding to C in *Figure 3E* (nucleotides G103-G107) is also difficult to resolve in both the T1 ladder and the probing assays. This region is base-paired in both *his* operator RNA conformations, thus it may also be difficult to disrupt and subsequently generate corresponding cleavage products.

Despite these limitations and although we were unable to identify any specific Hdp1$_{opt}$ binding site, the RNase T1 probing assays show that the *his* operator RNA indeed undergoes conformational changes in the presence of Hdp1$_{opt}$ and provide insight into the general regions of the *his* operator

RNA that are impacted by protein binding. Our data, as well as those from previous studies (*Digianantonio and Hecht, 2016*), indicate that the Hdps likely increase *hisB* expression by deattenuation of the *his* operator region. Therefore, it is possible that Hdp1$_{opt}$ binding results in changes to the *his* operator RNA secondary structure that favor the canonical anti-terminator conformation or promote the formation of an alternative anti-terminator structure. Sufficient stabilization of an anti-terminator conformation and/or destabilization of the *his* operator terminator upon Hdp binding would likely allow increased expression of *hisB* and ultimately the growth of the Δ*serB* auxotrophic mutant on minimal medium. Additional probing experiments are necessary to obtain more conclusive information on the RNA structural changes that take place upon Hdp1$_{opt}$ binding and how these changes may affect termination/anti-termination.

## Discussion

The recent identification of a multitude of small proteins encoded by sORFs raises intriguing questions about their functional roles and evolutionary origins. With regard to their emergence, one possible mechanism for generating sORFs is by degradation of larger genes into pseudogenes containing shorter coding regions (*Hemm et al., 2008*). Also in *E. coli*, other sORFs have been identified in prophage regions and thus may have origins stemming from an ancestral phage genomic integration event (*Hemm et al., 2008*; *VanOrsdel et al., 2018*; *Weaver et al., 2019*). However, these mechanisms only address the origins of a fraction of small protein-coding genes. A large majority of sORFs identified to date demonstrate little to no conservation between closely related phylogenetic classes and lack the genomic context to support or indicate either of the aforementioned mechanisms, suggesting that many of these small genes may have emerged de novo from previously non-genic sequences. Nonetheless, while previous studies show that novel functions, including variants with enzymatic activity, can be selected in bacteria from templates featuring localized randomization within larger functional progenitor sequences and/or structural scaffolds (*Digianantonio and Hecht, 2016*; *Hoegler and Hecht, 2016*; *Smith et al., 2015*), there is limited experimental evidence demonstrating that de novo genes can emerge from the expression of completely random and/or nonfunctional DNA (*Knopp et al., 2019*; *Knopp et al., 2021*). In this work, we identified de novo small proteins with RNA-binding regulatory activities.

We isolated three independent small proteins (Hdp1-Hdp3) that rescued an *E. coli* Δ*serB* mutant by upregulating the expression of an alternate enzyme, HisB, and for Hdp1 we showed direct binding interactions with the 5′ operator region of the *his* operon mRNA transcript. The *his* operon is regulated by a transcriptional attenuation mechanism (*Blasi and Bruni, 1981*; *Johnston et al., 1980*) as well as the stringent response that acts on the promoter and transcription initiation (*Riggs et al., 1986*; *Stephens et al., 1975*). Analysis of the β-galactosidase activity of several *his*$_{operator}$-*lacZ* transcriptional fusions in response to the presence of Hdp1 showed that (i) Hdp1 acts at the level of transcriptional attenuation (rather than the promoter), (ii) the stringent response is not involved in the Hdp1-mediated effects on *his* operon expression, and (iii) Hdp1 action requires the intact *his* operator region.

In addition, in vivo and in vitro experiments demonstrated that Hdp1 interacts with the *his* operator RNA and that protein binding results in changes to the RNA secondary structure that may facilitate increased transcription of the *his* operon, including *hisB*. Using a more soluble variant of Hdp1 (Hdp1$_{opt}$) that maintained the Δ*serB*-rescue phenotype, we showed by co-IP assays and EMSAs that Hdp1$_{opt}$ binds to the full-length *his* operator RNA with micromolar affinity and that the inactive protein mutant, Hdp1$_{opt}$ L27Q, has a fivefold reduction in affinity. Most RNA-protein regulatory interactions exhibit in vitro $K_d$ values within the nanomolar range or lower (*Ryder et al., 2008*). However, the moderate binding affinity of Hdp1$_{opt}$ to the *his* operator RNA is not surprising as the protein was directly selected from a random library without any further optimization.

More in-depth characterization of the binding interaction between Hdp1$_{opt}$ and the *his* operator RNA could be complicated by the possibility that the Hdps may exist as multimers rather than monomeric proteins as alpha-helical proteins with predominantly hydrophobic residues often have a propensity to oligomerize (*Li et al., 2012*). Hdp multimerization is conceivable as the semi-random proteins isolated from the previous Digianantonio and Hecht study that likely rescue Δ*serB* auxotrophy via a similar (but undetermined) mechanism of action were designed to fold into four-helix structures (*Digianantonio and Hecht, 2016*). Albeit beyond the scope of this work, further investigations

are needed to determine the precise molecular mechanisms underlying the binding model and/or stoichiometry of the Hdp-*his* RNA interaction.

As mentioned above, *Digianantonio and Hecht, 2016* also recovered several proteins (SynserB1-4) that rescued growth of a Δ*serB* auxotrophic mutant via HisB upregulation by utilizing libraries of partially randomized sequences. Even though the exact mechanism of the SynserB-mediated rescue is not characterized, it is likely that both the SynserBs and Hdps share a similar mechanism as both upregulate HisB expression in a stringent response-independent manner. Despite the functional similarity, there is no obvious sequence similarity between the SynserBs and Hdps, demonstrating that the selected function is not necessarily dependent on a specific extended sequence motif. That being said, the clustering of loss-of-function mutations in or around the predicted region of similarity within the Hdps is intriguing. However, we acknowledge that the small sample size, lack of overall sequence homology, and the short length of the proteins render it difficult to draw definitive conclusions on the functional relevance of this putative motif. As noted previously, additional experiments are required to confirm if the potential motif indeed plays any functional role in Hdp rescue activity.

Interestingly, we observed that expression of Hdp1 and Hdp3 causes a specific and significant upregulation of the *his* and *his*$_{operator}$-*lacZ* operons. In contrast, the SynSerB3 protein showed a rather unspecific effect on gene expression, with more than 600 proteins being affected (including most of the amino acid biosynthetic operons). It is possible that the differences in global proteome/transcriptome changes observed for *synserB3* and *hdp1* and *hdp3* are due to different experimental conditions. SynSerB3 was tested in a *serB*-deficient mutant in minimal medium, which results in severely reduced growth, whereas the Hdps were tested in rich medium, which allows for wild-type growth rates and comparisons with empty plasmid control strains, but admittedly does not perfectly recapitulate the conditions under which the Hdps were functionally selected. While determination of physiological effects upon Hdp expression under permissive conditions can still reveal a biological activity, it does not necessarily reflect the level of upregulation under selective conditions. This may be why we do not observe increased His and (modified) Lac protein abundance upon Hdp2 expression in our proteome analyses and why we measure a modest increase in β-galactosidase activity with Hdp2 in our reporter assays despite Hdp2's ability to rescue Δ*serB* strain growth on minimal medium.

Based on the binary pattern constraining the library design in the Digianantonio and Hecht study, the selected SynserB proteins form four-helix bundles. While this library design increases the chances of recovering well-structured proteins that likely reach high cellular concentrations without aggregating, bona fide de novo proteins evolved from completely random sequences do not share this privilege and will mostly form insoluble aggregates and be targeted for degradation (*Yomo et al., 1998*). Additionally, using libraries with preexisting structural properties further limits the selection of novel functionalities to those within the defined sequence and structural constraints. By utilizing expression libraries encoding completely random sORFs, we have shown that a new functionality can be selected truly de novo, without any predefined structural boundaries, demonstrating that preexisting structural scaffolds are not necessary for a novel protein to be expressed and achieve biological function with high specificity. However, randomized sequences in nature are likely not completely random as they may contain remnants of previously optimized structural or functional sequence motifs, which might increase the probability of acquiring a new function. Yet, the use of completely random sequences is a suitable starting point for proof-of-concept demonstration of the de novo evolution of small protein-coding genes as they serve as a true null model when screening for functionality, and unlike preexisting genomic sequences, they introduce little implicit bias or constraints during the selection process.

Given that the majority of small proteins described to date are predominately composed of hydrophobic amino acids (*Hemm et al., 2008*; *Vakirlis et al., 2020*), it is not surprising that all functional small proteins we have selected are also hydrophobic. As our previously recovered antibiotic resistance-conferring proteins are localized to the membrane, the strong hydrophobicity is linked to their functionality. However, despite the transmembrane domain predictions, it is unlikely that the Hdps exclusively associate with the cell membrane as their mechanism involves regulating transcription and the solubility-optimized variant of Hdp1 (Hdp1$_{opt}$) is not predicted to target the membrane (*Figure 1C*, *Figure 3—figure supplement 1*). Nevertheless, it is interesting to consider that hydrophobicity is an important contributor to functionality in random sequence space. This concept is supported by recent work from *Vakirlis et al., 2020*, which identified that the beneficial fitness

effects of emerging ORFs were found to be associated with the potential to produce transmembrane domains, and that thymine-rich intergenic regions in particular were identified as a reservoir for encoding transmembrane domains.

It is also worth noting that despite our extensive screen of 74 different auxotrophs, we only isolated positive 'hits' that rescued the ΔserB auxotrophic strain. As acquiring bona fide enzymatic activity from relatively short random DNA sequences is exceedingly rare, it is not unexpected that de novo-generated functionality primarily involves targeting preexisting cellular machinery and/or regulatory mechanisms. In this regard, the rescue of a ΔserB mutant is a convenient model system as *E. coli* already encodes a preexisting enzyme with a sufficient secondary activity (HisB) that can function in lieu of SerB when overexpressed, and *hisB* expression is modulated by a regulatory RNA structure (*his* operator region) that is an ideal target for interactions with novel small proteins.

The Hdps, as well as the small proteins selected from our past screens (*Knopp et al., 2019*; *Knopp et al., 2021*), demonstrate many qualities intrinsic to most known naturally occurring bacterial small proteins: (i) they are ≤50 amino acids in length, (ii) contain a high percentage of hydrophobic amino acids, (iii) are predicted to form predominantly alpha-helical structures, (iv) are predicted to associate with the membrane, and (v) they do not encode an enzymatic activity, but rather act as regulators/interactors to modulate a specific process (*Carvunis et al., 2012*; *Storz et al., 2014*). The previously isolated aminoglycoside resistance-conferring proteins (Arps) provide resistance by membrane insertion, disruption of the proton motive force, and a subsequent reduction in antibiotic uptake (*Knopp et al., 2019*), while the colistin resistance-conferring proteins (Dcrs) exert their effect via direct protein–protein interactions that activate the sensor kinase PmrB, resulting in Lipid A modifications and decreased affinity toward colistin (*Knopp et al., 2021*). This study extends the previously known spectrum of interacting partners of experimentally selected de novo small proteins to include nucleic acids and exemplifies how direct binding of a de novo protein to an RNA regulatory element can upregulate expression of a biosynthetic operon and restore growth of an auxotrophic *E. coli* strain. Directed evolution of our selected de novo small proteins could shed further light on the evolutionary constraints governing the emergence of new genes. For example, will selection fine-tune the function, expression, and/or stability of these proteins, and will these evolved variants converge to preexisting genes or differentiate to fill a novel and distinct structural and/or functional niche not yet occupied by naturally occurring genes?

## Materials and methods
### Strains
All strains were derivatives of *E. coli* MG1655 (F⁻, lambda⁻, *rph-1*). Gene deletion mutants were constructed by P1 transduction from the corresponding KEIO collection strains (derivatives of *E. coli* K-12 BW25113: F⁻, Δ(*araD-araB*)567, *lacZ4787*(del)::*rrnB-3*, LAM⁻, *rph-1*, Δ(*rhaD-rhaB*)568, *hsdR514*) (*Baba et al., 2006*). All *lacZ* gene fusions were constructed using *E. coli* NM580 and transferred to the test strains by P1 transduction (*Battesti et al., 2015*). For all strains generated in this study, aliquots (1 ml) from overnight cultures were cryopreserved with 10% DMSO and stored at –80°C. Where specified, strains were grown in/on lysogeny broth (LB Miller; 10 g/l NaCl, 10 g/l tryptone, 5 g/l yeast extract; Sigma-Aldrich), LB supplemented with 1.5% (w/v) agar (LA), Mueller-Hinton broth (MH; 17.5 g/l casein acid hydrolysate, 3 g/l beef extract, 1.5 g/l starch; BD Difco), or MH supplemented with 1.5% (w/v) agar (MHA). See *Supplementary file 1c* for a list of strains used and/or constructed for this study.

### Library construction
Expression vector libraries encoding 10, 20, and 50 amino acid-long sORFs were constructed as previously described (*Knopp et al., 2019*). Briefly, randomized oligonucleotides were complemented by primer extension and ligated into the expression vector pRD2 using BamHI and PstI restriction sites. Ligations were transformed into electrocompetent NEB5-alpha *E. coli* (New England Biolabs) and plated on LA plates containing 100 µg/ml ampicillin. After overnight incubation at 37°C, the cells were collected from the plate and plasmid pools were extracted using the NucleoBond Xtra Midi Kit (Macherey-Nagel) according to the manufacturer's recommendation.

## Selection of functional proteins

We selected 74 KEIO strains that were previously described to exhibit auxotrophic growth behavior (*Baba et al., 2006*; *Joyce et al., 2006*) including strains with a strong and weak (leaky) auxotrophic phenotype (see *Supplementary file 1a* for a complete list of the tested KEIO strains). An overnight culture of each single-gene knockout auxotroph was diluted 1:200 in 100 ml pre-warmed LB medium and incubated at 37°C and shaking at 200 rpm. When the cultures reached the target $OD_{600}$ of 0.2, flasks were quickly cooled in ice water for 10 min. The cells were pelleted by centrifugation at 4500 × *g* and 4°C and washed three times with cold 10% glycerol. Finally, cell pellets were resuspended in 300 µl 10% glycerol. Cells (40 µl) were mixed with 2 µl of each plasmid library (or empty vector control) in a chilled microcentrifuge tube and incubated on ice for 5 min. The mixture was transferred to an electroporation cuvette with a 1 mm gap width and transformed using a Gene Pulser Xcell electroporator (Bio-Rad) with 1.8 kV, 400 Ω, and 25 µF settings. The cells were recovered in 1 ml SOC medium (20 g/l tryptone, 5 g/l yeast extract, 0.5 g/l NaCl, 10 mM $MgCl_2$, 0.25 mM KCl, and 4 g/l glucose) for 1.5 hr at 37°C with shaking at 200 rpm. After recovery, the cells were washed twice with phosphate-buffered saline (PBS; 8 g/l NaCl, 0.2 g/l KCl, 1.44 g/l $Na_2HPO_4$, and 0.24 g/l $KH_2PO_4$) and resuspended in 1 ml PBS. An aliquot of each transformation (10 µl) was subjected to a dilution series, plated on LA supplemented with 100 µg/ml ampicillin, and incubated overnight at 37°C to determine transformation efficiencies. The remaining 990 µl were pelleted, washed twice with 1 ml PBS, resuspended in 200 µl PBS, and spread on M9 minimal medium plates (6 g/l $Na_2HPO_4$, 3 g/l $KH_2PO_4$, 0.5 g/l NaCl, 1 g/l $NH_4Cl$, 1 mM $MgSO_4$, 0.1 mM $CaCl_2$, 1.5% [w/v] agar) containing 0.2% glucose, 1 mM IPTG, and 100 µg/ml ampicillin. The plates were incubated for up to 14 days at 37°C and regularly inspected for growth of colonies.

When the empty vector control transformations showed growth of a faint lawn, plates containing this auxotrophic strain were discarded. Colonies that appeared during the selection were re-streaked on M9 minimal medium plates containing 0.2% glucose, 1 mM IPTG, and 100 µg/ml ampicillin, and the original plate was incubated further to allow the growth of additional slower-growing colonies. Plasmids were isolated from overnight cultures inoculated with the re-streaked colonies using the EZNA Plasmid Mini Kit (Omega Bio-Tek) and reintroduced into the parental plasmid-free auxotrophic strain to confirm whether the rescue was plasmid-mediated or due to chromosomal mutations. Plasmids that were confirmed to mediate rescue of the auxotrophic *E. coli* mutant were sequenced, and the insert was re-synthesized and cloned into the empty pRD2 vector to confirm that the insert causes the rescue rather than any alterations on the plasmid. In the cases where all experiments indicated a rescue mediated by the insert, further mechanistic characterizations were conducted.

## Plasmid transformation

For transformation or re-transformation of miniprepped plasmids, 2 ml of an overnight culture were washed three times with 10% glycerol and the pellet was resuspended in 200 µl 10% glycerol. Cells (40 µl) were mixed with 200 ng of the desired plasmid and incubated on ice for 5 min. Transformation via electroporation was performed as described above and cells were recovered in 1 ml pre-warmed MH for 1 hr at 37°C with shaking at 200 rpm. Cells were then plated on MHA with 100 µg/ml ampicillin and incubated overnight at 37°C and select transformants were subsequently purified in an additional re-streak. See *Supplementary file 1f* for a list of all plasmids used in this study.

## Sequence analysis

Local sequencing analyses were performed using CLC Main Workbench (QIAGEN). Secondary structure predictions were performed using AlphaFold (*Jumper et al., 2021*; *Mirdita et al., 2022*) and JPred4 (*Drozdetskiy et al., 2015*). Multiple sequence analyses were performed using Clustal Omega (*Sievers et al., 2011*) using standard parameters. Hydropathy scores were determined using the integrated CLC tool applying the Kyte-Doolittle hydrophobicity scale and a window size of 11.

## Construction of protein variants

Variants of the isolated inserts were either ordered as ready-made constructs from Geneart (Thermo Fisher) subcloned into pRD2 or constructed by cloning of annealed oligos. For the latter, two complementary oligonucleotides (Eurofins Genomics) containing single-stranded ends corresponding to BamHI and PstI cleavage sites were combined with annealing buffer (10 mM Tris pH 7.5, 100 mM

NaCl, 100 µM EDTA) at equimolar ratios to a final concentration of 1 µM in a microcentrifuge tube and transferred to 95°C hot water, which was then slowly cooled to room temperature. The resulting double-stranded DNA was then ligated into the BamHI/PstI-digested pRD2 vector using Ready-To-Go T4 DNA Ligase (GE Healthcare Life Sciences) according to the manufacturer's recommendation. The ligation reaction was purified using the GeneJET Gel Extraction Micro Kit (Thermo Fisher). To transform the final construct, an overnight culture of *E. coli* NEB5-alpha (New England Biolabs) was diluted 1:100 in LB medium and grown to $OD_{600}$ of 0.2. The cells were then cooled on ice for 10 min and washed three times with cold 10% glycerol, 40 µl of cells were mixed with 2 µl of each ligation reaction, and transformed via electroporation as described above. Cells were recovered in 1 ml pre-warmed MH for 1 hr at 37°C with shaking at 200 rpm. Transformants were selected on LA plates containing the appropriate antibiotics. Plasmids were isolated using the EZNA Plasmid Mini Kit (Omega Bio-Tek) and verified via sequencing. Confirmed constructs were transformed into the strains of interest as described above.

## Site-directed mutagenesis

PCR-amplification of the entire plasmid containing the gene of interest was performed with Phusion DNA polymerase (Thermo Fisher) using two complementary oligonucleotides containing a stretch of overlapping bases with the desired mutation in the middle (*Supplementary file 1g*). The reaction product was purified using GeneJET Gel Extraction Kit (Thermo Fisher) and digested with DpnI (Thermo Fisher) for 1 hr at 37°C to remove the template DNA. After digestion, the PCR product was purified and transformed into NEB 5-alpha electrocompetent *E. coli* (New England Biolabs) as described above, plated on LA plates containing 50 µg/ml of ampicillin, and incubated overnight at 37°C. Transformants were inoculated into fresh LB medium supplemented with 50 µg/ml of ampicillin and grown overnight at 37°C with shaking at 200 rpm. Plasmids were purified from the overnight cultures using the EZNA Plasmid Mini Kit (Omega Bio-Tek) according to the manufacturer's protocol. Presence of the desired nucleotide substitutions were confirmed by Sanger sequencing.

## Random mutagenesis

Random mutagenesis of the *hdp1* and *hdp2* genes was performed with the GeneMorph II Random Mutagenesis Kit (Agilent) using the recommended conditions for obtaining a mean mutation rate of 1 mutation per PCR product. The resulting PCR product was used as a megaprimer for whole-plasmid PCR using Phusion DNA polymerase (Thermo Fisher) (*Sarkar and Sommer, 1990*), which was then purified using the GeneJET PCR Gel Extraction Kit (Thermo Fisher) and digested with DpnI (Thermo Fisher) for 1 hr at 37°C to remove the original template plasmid. After digestion, the PCR product was purified again and transformed into electrocompetent DA57390 cells (*Supplementary file 1c*), which were plated on MacConkey (BD Difco) plates supplemented with 10% lactose and 50 µg/ml of ampicillin and incubated overnight at 37°C. White colonies indicated loss-of-function mutations in the mutagenized genes. These colonies were inoculated into the fresh LB medium supplemented with 50 µg/ml ampicillin and grown overnight at 37°C with shaking at 200 rpm. The plasmids were purified using the EZNA Plasmid Mini Kit (Omega Bio-Tek) according to the manufacturer's protocol. The nucleotide substitutions were identified by Sanger sequencing.

## Growth assays

Growth curves were measured using a Bioscreen C (Oy Growth Curves AB, Ltd., Finland). Overnight cultures of select strains were grown in triplicate in LB broth supplemented with 50 µg/ml ampicillin at 37°C with shaking at 200 rpm, and diluted 1:1000 into 1 ml fresh LB medium supplemented with 50 µg/ml ampicillin ±1 mM IPTG. From this dilution, two 300 µl aliquots were transferred into Bioscreen C honeycomb well plates (for technical replicates). Plates were incubated in the Bioscreen C for 24 hr at 37°C with continuous shaking and the $OD_{600}$ was measured every 4 min. The background $OD_{600}$ of wells containing blank un-inoculated media were subtracted from the appropriate/corresponding $OD_{600}$ growth measurements and growth rates were calculated from the linear slope of $\ln(OD_{600})$ in exponential phase ($\ln(OD_{600})$ -4 to -2). Relative growth rates were calculated by dividing each growth rate by the growth rate of the appropriate empty plasmid strain from the corresponding condition. The values reported represent the mean of three independent biological replicates, with two technical replicates for each biological replicate; error reported is the standard deviation.

## Chromosomal *lacZ* fusions

Chromosomal fusions to the *lacZ* reporter gene were generated using the lambda-red recombination system described in *Battesti et al., 2015*. A full list of the *lacZ* fusion genetic constructs and their sequences is shown in *Supplementary file 1c*. In this system, *E. coli* strain NM580 carries the temperature-inducible lambda *red* gene, which is used for a standard recombineering technique (*Sharan et al., 2009*). The sequence of interest replaces the arabinose-inducible *ccdB* toxin gene and adjacent kanamycin-resistance gene upstream of *lacZ*. A zeocin resistance cassette located upstream of the fusion was used for the selection of P1 transductants.

*E. coli* strain NM580 was streaked from the frozen stock on LA plates supplemented with 1% glucose and 50 µg/ml of kanamycin and grown at 30°C overnight. One colony from the plate was inoculated into 3 ml of fresh LB broth supplemented with 1% glucose and 50 µg/ml kanamycin and grown at 30°C with shaking at 200 rpm until an OD$_{600}$ of 0.2 was reached. The cultures were transferred to a 43°C water bath, incubated for 20 min with shaking, and cooled on ice. Cells were collected and washed three times with 20 ml of ice-cold 10% glycerol. The cells were pelleted and resuspended in 200 µl of ice-cold deionized water. Aliquots (50 µl) were used for electroporation with the corresponding PCR product (in 0.5 µl of water) as described above. Cells were recovered in 1 ml of LB with 1% glucose for 1 hr at 37°C with shaking at 200 rpm. Cells were then pelleted and plated on LA supplemented with 0.2% arabinose and 20 µg/ml zeocin. After overnight incubation at 37°C, kanamycin-susceptible colonies were confirmed with PCR and Sanger sequencing.

PCR products for the recombineering were designed to have 40 base overlaps with the chromosomal region necessary for recombination and were amplified using Phusion DNA polymerase (Thermo Fisher). When necessary, two-step megaprimer PCR was used, wherein the product synthesized in the first step served as a primer in the second step (*Sarkar and Sommer, 1990*). Oligonucleotides used are listed in *Supplementary file 1g*. P1 phage transductions were used to transfer the *lacZ* reporter constructs generated in *E. coli* NM580 to the test strains of interest according to published methods (*Battesti et al., 2015*).

## β-Galactosidase activity assays

Overnight cultures were grown in triplicate in LB broth supplemented with 50 µg/ml ampicillin at 37°C, and 300 µl of each culture was used to inoculate 3 ml of fresh LB medium supplemented with 1 mM IPTG and 50 µg/ml ampicillin. For the arabinose-inducible P$_{araBAD}$ reporter strain, 0.2% of arabinose was also added to the medium. Cells were grown to early stationary phase (OD$_{600}$ 1.0–2.2), harvested, and β-galactosidase activity was assayed and measured using a Bioscreen C machine (Oy Growth Curves AB, Ltd, Finland) essentially as described in *Kacar et al., 2017*. Reaction time course was measured at 28°C and 420 nm wavelength at 1 min intervals without shaking and β-galactosidase activity in Miller Units was calculated using the following formula, where t2 and t1 are times of the measurements in the linear range of the reaction: $1000*\frac{(OD_{420,t2}-OD_{420,\,t1})}{OD_{600}*0.2*(t2-t1)}$ (modified from *Zhang and Bremer, 1995*). The values reported represent the mean of three or more independent biological replicates; error bars represent the standard deviation.

## Sample preparation for total proteome analysis

For the proteomic analysis, cells expressing either the empty vector control, *hdp1*, *hdp2*, *hdp3*, or *hdp1 L27Q* were grown until early stationary phase (OD$_{600}$ = 1.4–1.6) in LB medium in the presence of 1 mM IPTG and 50 µg/ml ampicillin. Cells (5 ml) were collected and cell weight was determined. Samples were prepared and an equal amount of each sample was separated on an SDS-PAGE until the dye front reached the bottom of the gel. The whole lane from each sample was cut out from the gel for further analysis. Experiments were performed with three independent biological replicates for each strain.

## Label-free proteomic quantification

Each gel lane was cut into two or six separate pieces, and proteins were reduced in-gel with 10 mM DTT in 25 mM NH$_4$HCO$_3$, alkylated with 55 mM iodoacetamide in 25 mM NH$_4$HCO$_3$, and thereafter digested with 17 ng/µl sequencing-grade trypsin (Promega) in 25 mM NH$_4$HCO$_3$ using a slightly modified in-gel digestion protocol (*Shevchenko et al., 1996*). The resulting peptides were then eluted

from the gel pieces using 1% (v/v) formic acid (FA) in 60% (v/v) acetonitrile, dried down in a vacuum centrifuge (ThermoSavant SPD SpeedVac, Thermo Scientific), and finally dissolved in 1% (v/v) FA.

## Liquid chromatography and mass spectrometry

Peptide samples were desalted using Stage Tips (Thermo Fisher) according to the manufacturer's protocol and subsequently dissolved in 0.1% (v/v) FA (solvent A). Desalted samples were separated by RP-HPLC using a Thermo Scientific nLC-1000 with a two-column setup, where an Acclaim PepMap 100 (2 cm × 75 µm, 3 µm particles; Thermo Fisher) pre-column was connected in front of an EASY-Spray PepMap RSLC C18 reversed phase column (50 cm × 75 µm, 2 µm particles; Thermo Fisher). The column was heated to 35°C and equilibrated in solvent A. A gradient of 2–40% solvent B (acetonitrile and 0.1% [v/v] FA) was run at 250 nl/min for 3 hr. The eluted peptides were analyzed on a Thermo Scientific Orbitrap Fusion Tribrid mass spectrometer, operated at a Top Speed data-dependent acquisition scan mode, ion-transfer tube temperature of 275°C, and a spray voltage of 2.4 kV. Full-scan MS spectra (m/z 400–2000) were acquired in profile mode at a resolution of 120,000 at m/z 200 and analyzed in the Orbitrap with an automatic gain control (AGC) target of 2.0e5 and a maximum injection time of 100 ms. Ions with an intensity above 5.0e3 were selected for collision-induced dissociation (CID) fragmentation in the linear ion trap at a collision energy of 30%. The linear ion trap AGC target was set at 1.0e4 with a maximum injection time of 40 ms, and data was collected at centroid mode. Dynamic exclusion was set at 60 s after the first MS1 of the peptide. The system was controlled by Xcalibur software (version 3.0.63.3; Thermo Scientific). Quality control of the instrument was monitored using the Promega 6x5 LC-MS/MS Peptide Reference Mix before and after each MS experiment run, and analyzed using PReMiS software (version 1.0.5.1, Promega).

## Mass spectrometric data analysis

Data analysis of raw files was performed using MaxQuant software (version 1.6.2.3) and the Andromeda search engine (*Cox et al., 2009*; *Tyanova et al., 2016*), with cysteine carbamidomethylation as a static modification and methionine oxidation and protein N-terminal acetylation as variable modifications. First search peptide MS1 Orbitrap tolerance was set to 20 ppm and ion trap MS/MS tolerance was set to 0.5 Da. Match between runs was enabled to identify peptides in fractions where only MS1 data were available. Minimum LFQ ratio count was set to 2, and the advanced ratio estimation option was enabled. Peak lists were searched against the UniProtKB/Swiss-Prot *Escherichia coli* K12 proteome database (UP000000625, version 2019-03-27), including the Hdp1 protein sequence, with a maximum of two trypsin miscleavages per peptide. The contaminants database of MaxQuant was also utilized. A decoy search was made against the reversed database, where the peptide and protein false discovery rates were both set to 1%. Only proteins identified with at least two peptides of at least seven amino acids in length were considered reliable. The peptide output from MaxQuant was filtered by removing reverse database hits, potential contaminants and proteins only identified by site (PTMs). Differential expression analysis was performed by the DEP 1.7.0 package for Bioconductor and R (*Zhang et al., 2018*). The LFQ intensity data was normalized by the variance stabilizing transformation (vsn) method, and missing values were imputed by a maximum likelihood-based imputation method using the EM algorithm (*Gatto and Lilley, 2012*). Protein-wise linear models and empirical Bayes statistics using LIMMA were used for the differential expression calculation (*Ritchie et al., 2015*). The p-values were adjusted for multiple testing using the Benjamini–Hochberg method (*Benjamini and Hochberg, 1995*). The mass spectrometry proteomics data have been deposited to the ProteomeXchange Consortium (http://proteomecentral.proteomexchange.org) via the PRIDE partner repository (*Perez-Riverol et al., 2019*) with the dataset identifiers PXD014049 and PXD040161.

## Co-immunoprecipitation assays

Select *E. coli* Δ*serB* strains with the full-length $P_{hisL}$-$his_{operator}$-*lacZ* reporter and pRD2 containing either Hdp1$_{opt(+cys)}$-HA tag, Hdp1$_{opt(+cys)}$ L27Q-HA tag, or untagged Hdp1$_{opt(+cys)}$, were grown from single colonies in 2 ml LB supplemented with 50 µg/ml ampicillin overnight at 37°C with shaking at 200 rpm. These cultures were used to inoculate 100 ml LB cultures supplemented with 50 µg/ml ampicillin and 1 mM IPTG, which were then grown at 37°C with shaking at 200 rpm until an $OD_{600}$ of ~2.0 was reached. Two 50 ml aliquots of each culture were cooled on ice, washed twice with 1× PBS, and cell pellets were stored at –80°C until use.

For the co-immunoprecipitation experiments, cell pellets were thawed on ice, resuspended in 300 μl B-PER II Bacterial Protein Extraction Reagent (Thermo Scientific), and incubated at room temperature for 15 min with occasional vortexing. Cell lysates were clarified by centrifugation at 15,000 × $g$ for 5 min at 4°C, supernatants were transferred to new tubes, and two 50 μl aliquots of each lysate were saved for RNA extractions/RT-qPCR and protein precipitations/Western blots. Anti-HA Magnetic Beads (Pierce, Thermo Scientific) were washed using IP Lysis/Wash Buffer (Pierce, Thermo Scientific) according to the manufacturer's protocol and the remaining lysates were added to the beads and incubated on a rocker at 4°C for 2 hr. After incubation, the unbound lysate fractions were removed from the beads using a magnetic stand, 50 μl aliquots of each 'unbound fraction' were saved for western blot analysis, and the magnetic beads were washed three times with 1 ml 1× Tris-buffered saline (TBS; 6.05 g/l Tris, 8.76 g/l NaCl, pH 7.5). After the final wash, 50 μl aliquots of each 'wash fraction' were saved for western blot analysis. Bound proteins of interest were eluted from the beads with the addition of 100 μl of 20 mM Tris-HCl pH 7.5 +1% SDS, followed by incubation at 70°C for 5 min with occasional vortexing. The eluted fractions were recovered from each sample and divided into two 50 μl aliquots; one for RNA extractions/RT-qPCR and one for protein precipitations/western blot analysis.

## Western blots

Saved protein fractions from the different steps of the co-immunoprecipitation experiments were acetone precipitated, resuspended in 1× Tricine sample buffer (Bio-Rad) (150 μl for lysate/input and unbound fractions, 30 μl for wash and eluate/output fractions), incubated at 65°C for 4 min with occasional vortexing, and cooled on ice. Aliquots (10 μl) of each fraction were then incubated at 95°C for 4 min and separated on a 16.5% Tris-Tricine Mini-PROTEAN gel (Bio-Rad) in 1× Tris-Tricine Running Buffer (Bio-Rad) at 100 V until the dye front was approximately 1 cm from the bottom of the gel. Protein samples were transferred to a PVDF membrane using a Trans-Blot Turbo Mini PVDF Transfer Pack and Trans-Blot Turbo Transfer System (Bio-Rad) with the 'Mixed MW' program (2.5 A, 25 V, 7 min). Membranes were blocked overnight at 4°C with rocking with 12 ml TBS + 1% Tween 20 (TBS-T) + 5% milk, washed once with 12 ml TBS-T for 10 min at room temperature with rocking, and then probed with 1:3000 dilution of HRP-conjugated anti-HA mouse monoclonal antibody (Invitrogen) (4 μl) in 12 ml TBS-T + 3% bovine serum albumin (BSA; Merck) for 1 hr at room temperature with rocking. Membranes were then washed three times with 12 ml TBS-T for 10 min at room temperature with rocking. Detection of tagged proteins of interest was performed using Amersham ECL Western Blotting Detection Reagent (Cytiva) according to the manufacturer's protocol, and the membranes were visualized using a Bio-Rad ChemiDoc MP system (Chemi Hi Sensitivity setting).

## RT-qPCR

RNA was isolated from the appropriate fractions from the different steps of the co-immunoprecipitation experiments via phenol-chloroform extraction followed by ethanol precipitation. Genomic DNA was digested from the RNA samples using the TURBO DNA-free Kit (Invitrogen) and reverse-transcription was performed with 150 ng of each DNase-treated RNA sample using the High Capacity cDNA Reverse Transcription kit (Applied Biosystems). RNA transcripts of interest were detected via qPCR using the resulting cDNA, PerfeCTa SYBR Green FastMix (QuantaBio), and an Illumina Eco Real-Time PCR machine. qPCR was performed with RNA samples from three independent biological replicates, with three technical replicates for each biological replicate (for oligos used, see **Supplementary file 1g**). Experiments were also repeated with template samples lacking reverse transcriptase to confirm the effective removal of genomic DNA.

To determine fold-enrichment of RNA transcripts, the $\Delta C_T$ for each target transcript from both the lysate/input and eluate/output RNA fractions was calculated by subtracting the mean $C_T$ value measured from the appropriate untagged Hdp1$_{opt(+cys)}$ sample from that of the corresponding HA-tagged protein (WT or L27Q): $\Delta C_T = C_{T\ HA\text{-tagged}} - C_{T\ untagged}$. Fold-enrichment was then calculated as: Fold-enrichment $= \dfrac{2^{-\left(\Delta C_{T\ output}\right)}}{2^{-\left(\Delta C_{T\ input}\right)}}$. Values reported represent the mean of three independent biological replicates, with three technical replicates for each biological replicate; error bars represent the standard deviation.

## Electrophoretic mobility shift assays (EMSAs)

The region corresponding to the full-length *his* or *thr* operator sequence was PCR-amplified from *E. coli* MG1655 genomic DNA using a forward primer that contained the T7 promoter sequence (**Supplementary file 1g**). The resulting PCR product was column-purified using the GeneJet Gel Extraction Kit (Thermo Fisher), and RNA was transcribed from this construct (0.2 µg template) using the MEGAscript T7 RNA Polymerase Kit (Life Technologies) according to the manufacturer's protocol, with overnight incubation. The RNA was DNase-treated using the TURBO DNA-free Kit (Invitrogen), and gel purified by 6% denaturing PAGE (**Milligan et al., 1987**). The purified RNA (60 pmol) was then dephosphorylated using CIAP (Thermo Fisher), phenol-chloroform extracted and ethanol precipitated, and 5 pmol was 5'-end labeled with 20 µCi of γ-P$^{32}$ ATP (Perkin Elmer) using Polynucleotide Kinase (Thermo Fisher) and subsequently purified using an Illustra MicroSpin G-50 column (Cytiva). The Hdp1$_{opt}$ and Hdp1$_{opt}$ L27Q proteins used for the in vitro experiments were synthesized by GenScript and resuspended in water for a stock concentration of 1 mg/ml.

For the binding reactions, a twofold dilution series was prepared for each protein in 2× binding buffer (50 mM Tris-HCl pH 7.5, 200 mM NaCl, 2 mM MgCl$_2$), and 2.5 fmol of labeled RNA diluted in water was denatured at 95°C for 1 min, cooled on ice for 2 min, and combined with an equal volume of 2× binding buffer. The RNA (5 µl) was then combined with the protein serial dilutions for a total reaction volume of 10 µl in 1× binding buffer (resulting in a final RNA concentration of 1.25 nM/reaction and a final protein concentration range of approximately 0–5.5 µM). Reactions were incubated at 37°C for 15 min and then 5 µl of loading buffer (48% glycerol, 0.01% bromophenol blue) was added to each reaction, and the samples were separated on a native 5% acrylamide gel run at 200 V at 4°C for 3 hr with 0.5× TBE buffer (50 mM Tris, 50 mM boric acid, 1 mM EDTA pH 8.0). The gels were exposed to a phosphor screen overnight at –20°C and visualized using a Bio-Rad Personal Molecular Imager.

Band intensity was quantified using Bio-Rad Image Lab Software 6. The fraction-bound RNA was calculated from the band intensities as bound/(bound + unbound). Fraction-bound RNA was then plotted versus concentration of Hdp1$_{opt}$ protein and a hyperbolic equation derived from a one-site binding model was fitted to the data to estimate $K_d$ values: Fraction bound = $B_{max}$ [Hdp1$_{opt}$]/( $K_d$ + [Hdp1$_{opt}$])+$C$, where $B_{max}$ represents the maximum fraction bound, $K_d$ the dissociation constant, and $C$ the observed fraction bound at 0 µM Hdp1$_{opt}$. Alternatively, a model accounting for positive cooperativity was also fitted to the data: Fraction bound = $B_{max}$ [Hdp1$_{opt}$]$^h$/( $K_d^h$ + [Hdp1$_{opt}$]$^h$)+$C$, where $h$ is the Hill coefficient. The latter equation accounts for a model in which two Hdp1$_{opt}$ proteins can bind to one RNA molecule, and where the affinity for the second Hdp1$_{opt}$ is higher than that for the first one. Curve fitting was performed with Prism 9 (GraphPad Software). Refer to **Supplementary file 1d and e** for the quantified EMSA data and the fitted parameters for each model. EMSAs were repeated independently three or more times for each protein; representative gels are shown (**Figure 3—figure supplement 3**). The quantified values reported represent the mean of three or more independent experimental replicates; the error bars represent the standard error.

## RNase T1 probing assays

Binding reactions with the radiolabeled full-length *his* operator RNA and the Hdp1$_{opt}$ protein were carried out as described above (using final protein concentrations of 0 and 5.5 µM). Following incubation, 0.05 U of RNase T1 (Ambion) was added, and the reactions were incubated at 37°C for an additional 10 min. Cleavage was stopped with the addition of 5.5 µl 0.1 M EDTA pH 8.0 (0.33 mM final concentration), and RNA fragments were recovered via ethanol precipitation and resuspended in 10 µl water and 10 µl Gel Loading Buffer II (Ambion). The OH ladder was generated by incubating the RNA in 1× Alkaline Hydrolysis Buffer (Ambion) at 95°C for 12 min. To generate the denaturing T1 ladder, the RNA was combined with 1× Sequencing Buffer (Ambion), denatured at 95°C for 1 min, cooled on ice, and then incubated with 0.1 U RNase T1 (Ambion) at 37°C for 5 min. Following incubation, ladder reactions were combined with an equal volume of Gel Loading Buffer II (Ambion) and kept on ice. Prior to gel electrophoresis, all samples were denatured at 95°C for 1 min, cooled on ice, and then 5 µl were loaded on an 8% denaturing polyacrylamide gel and run at 30 W at room temperature with 1× TBE (100 mM Tris, 100 mM boric acid, 2 mM EDTA pH 8.0). Gels were dried, exposed to phosphor screens for 24–48 hr, and visualized using a Bio-Rad Personal Molecular Imager. RNase T1 probing assays were repeated independently three times with similar results; a representative gel is shown in the main text (**Figure 3D**, **Figure 3—figure supplement 4**).

## Acknowledgements

This work was supported by grants from the Wallenberg Foundation (grant 2015.0069 to DIA, grant 2017.0071 to Leif Andersson for the proteomics work) and the Swedish Research Council (grant 2017-01527 to DIA, grant 2019-00666 to MK, grant 2020-04395 to PJ). The funders had no role in study design, data collection and analysis, decision to publish, or preparation of the manuscript. The authors would like to thank Michelle Meyer, Gerhart Wagner, and Omar Warsi for the helpful feedback and suggestions, Roderich Römhild for his help with the growth curve analysis, and Cedric Romilly and Thomas Stenum for their assistance with the in vitro experiments.

## Additional information

### Funding

| Funder | Grant reference number | Author |
| --- | --- | --- |
| Knut och Alice Wallenbergs Stiftelse | 2015.0069 | Dan I Andersson |
| Vetenskapsrådet | 2017-01527 | Dan I Andersson |
| Vetenskapsrådet | 2019-00666 | Michael Knopp |
| Vetenskapsrådet | 2020-04395 | Per Jemth |
| Knut och Alice Wallenbergs Stiftelse | 2017.0071 | Mårten Larsson |

The funders had no role in study design, data collection and interpretation, or the decision to submit the work for publication.

### Author contributions

Arianne M Babina, Formal analysis, Investigation, Visualization, Methodology, Writing – original draft, Writing – review and editing; Serhiy Surkov, Formal analysis, Validation, Investigation; Weihua Ye, Investigation; Jon Jerlström-Hultqvist, Formal analysis, Investigation; Mårten Larsson, Data curation, Formal analysis, Investigation; Erik Holmqvist, Formal analysis, Methodology; Per Jemth, Supervision, Funding acquisition, Formal analysis; Dan I Andersson, Conceptualization, Supervision, Funding acquisition, Writing – review and editing; Michael Knopp, Conceptualization, Formal analysis, Funding acquisition, Investigation, Visualization, Methodology, Writing – original draft, Writing – review and editing

### Author ORCIDs

Arianne M Babina http://orcid.org/0000-0002-4635-8396
Per Jemth http://orcid.org/0000-0003-1516-7228
Dan I Andersson http://orcid.org/0000-0001-6640-2174
Michael Knopp http://orcid.org/0000-0002-8218-3263

### Decision letter and Author response

Decision letter https://doi.org/10.7554/eLife.78299.sa1
Author response https://doi.org/10.7554/eLife.78299.sa2

## Additional files

### Supplementary files

• Supplementary file 1. Supplementary Tables 1a-g. (a) KEIO deletion strains screened with the random sequence libraries. (b) Summary of Hdp1-2 single mutants obtained via random mutagenesis. (c) *Escherichia coli* K-12 strains used in this study. (d) Fraction bound data as quantified and calculated from the electrophoretic mobility shift assay (EMSA) gels. (e) Parameters from fitting of different binding models to the EMSA data. (f) Plasmids used in this study. (g) Oligonucleotides used in this study.

• MDAR checklist

## Data availability

All data generated or analysed during this study are included in the manuscript and supporting file. The mass spectrometry proteomics data have been deposited to the ProteomeXchange Consortium (http://proteomecentral.proteomexchange.org) via the PRIDE partner repository (*Perez-Riverol et al., 2019*) with the dataset identifiers PXD014049 and PXD040161.

The following datasets were generated:

| Author(s) | Year | Dataset title | Dataset URL | Database and Identifier |
|---|---|---|---|---|
| Larsson M | 2022 | Mass spectrometry proteomics data | https://www.ebi.ac.uk/pride/archive/projects/PXD014049 | PRIDE, PXD014049 |
| Larsson M | 2023 | Mass spectrometry proteomics data | https://www.ebi.ac.uk/pride/archive/projects/PXD040161 | PRIDE, PXD040161 |

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
