## [Editor Report]

This important study shows that small proteins encoded by randomized DNA sequences can be biologically active in bacteria, suggesting an evolutionary pathway for the creation of new biological functions. The authors use a combination of genetic and biochemical approaches to convincingly demonstrate a regulatory function for a randomly generated small protein in *Escherichia coli*. The work will be of interest to scientists working in the field of molecular evolution and cellular innovation.

---

## [Decision Letter]

**Decision letter after peer review:**

Thank you for submitting your article "Rescue of *Escherichia coli* auxotrophy by de novo small proteins" for consideration by *eLife*. Your article has been reviewed by 3 peer reviewers, including Joseph T Wade as Reviewing Editor and Reviewer #1, and the evaluation has been overseen by Philip Cole as the Senior Editor. The following individual involved in the review of your submission has agreed to reveal their identity: Mona Wu Orr (Reviewer #3).

The reviewers found the paper to be interesting, and all reviewers agreed that the model for Hdp1 function is exciting since it would demonstrate a regulatory function associated with a newly evolved small protein. However, the reviewers also felt that the paper requires additional work to convincingly show that Hdp1 functions by specifically binding the his mRNA 5' UTR. The reviewers have put together a list of essential revisions. Most of these should be straightforward. The one new experiment the reviewers felt is important is to demonstrate a specific interaction of Hdp1 with the his mRNA 5' UTR in vivo.

Essential revisions:

1. Add a table with a list of the Hdp1 and Hdp2 random mutants assayed, including any that retained function.

2. Soften the conclusions regarding the putative shared amino acid motif (Figure 1D).

3. Include all relevant bacterial growth data.

4. Include a negative control RNA in the EMSA experiment.

5. Add reporter assay data for Hdp1opt and Hdp1opt L27Q using one of the lacZ reporter fusions, to establish whether the L27Q mutation in the context of Hdp1opt disrupts function.

6. Add a replicate of the RNase T1 experiment; quantify the data.

7. Demonstrate a specific interaction of Hdp1 (or Hdp1opt) with the his mRNA 5' UTR in vivo. If this involves a tagged Hdp1, include controls to show that the tagged protein retains activity in vivo.

Please also respond to comments made by individual reviewers.

*Reviewer #1 (Recommendations for the authors):*

1. The authors describe several results involving the growth of different strains, e.g. growth rescue of the serB deletion by the three small proteins, and growth of hisB/serA/serC mutants. These data should be shown.

2. The authors describe random mutagenesis of Hdp1 and Hdp2. A full list of the identified mutations should be included as a table. Were any mutants identified and sequenced that maintained function? If so, these should be included in the table.

3. The motif shown in Figure 1D may be important, but the evidence supporting this idea is relatively weak. The sequences are somewhat divergent between the three proteins, and the previously reported protein SynSerB3 lacks a match to this motif. Conclusions about the importance of the motif should be softened.

4. The Hdp1opt L27Q mutant should be tested for its effect in vivo with the reporter assay.

*Reviewer #2 (Recommendations for the authors):*

– Currently, all growth/fitness data are not presented. Growth curves or spot plate assays for all relevant experiments should be added, among which for Hpd1-3 random proteins, Hpd1-3 versions with premature stop codons or truncations, recoded Hdp ORFs, Hdp1(L27Q), Hdp1-optimized, Hdp1-optimized(L27Q), claim that cells expressing Hdp2 grow slower than cells expressing Hdp1 or Hdp3, experiments in ΔserA, ΔserC, ΔYtjC, Δgph, or ΔhisB backgrounds.

These experiments should include the following controls: (i) an empty vector, (ii) a SerB complementation on the same vector, and (iii) a HisB complementation on the same vector.

The fitness of cells expressing Hpd1-3 in medium supplemented with serine should also be included to show their cost/toxicity.

– The authors are encouraged to clearly state the model for Hdps molecular function before describing the data in figure 2 to help the readers evaluate their findings. If I understand correctly, the model suggests that when histidine-charges tRNA levels are high, there are low transcript levels of the his operon. Expression of Hdps results in increased RNA levels of the his operon, leading to increased protein levels of HisB that compensate for the SerB deletion. To test this model the authors should:

– Demonstrate that in a medium without histidine and serine, HisB protein levels are high enough to allow the growth of ΔserB cells.

– Demonstrate that in a medium without histidine both the fitness improvement and the increase in HisB protein levels as a result of Hdps expression are dramatically reduced because HisB levels are already high due to activation of the his operon and the effects of Hdps are partially/completely alleviated.

– It is unclear if LacZ levels are statistically significant between an empty vector and Hdp2 expression. The authors should clarify this issue and also directly measure HisB protein levels upon Hdp2 expression. This is critical to support the claim that Hdp2 functions similarly to Hdp1 and Hdp3.

– LacZ levels for cells expressing Hdp1(L27Q) seem very similar to LacZ levels for cells expressing Hdp2, however, the authors claim that the former is non-functional and the latter is functional. The authors should show growth data, provide statistical analysis for this experiment, and directly measure HisB protein levels for each construct. This is critical for the claim that increasing HisB levels is the mechanism of rescue by Hdps.

– Figure 2E does not support the authors' claim because constructs (ii)-(iv) show high basal expression levels. Therefore, any Hdp1 effect might be negligible in this context. The authors should find mutations in the his operator that maintain its functionality on one hand but alleviate Hdp1's function. Because this claim is central to the model presented in the paper, this control is very important.

– The thr-lacZ construct is not a good control for the experiment in figure 2B because its basal LacZ levels are high in the conditions of the experiment (similar to my previous comment). Yet, this is a critical control, so the authors should find a different operator with a similar structure and function to the his operator and for which LacZ basal levels are low for an empty vector control. If LacZ levels remain low upon Hdp1 expression, that will be an important support for the authors' model.

– Such an alternative operator control should be used as the control for the EMSA experiments and T1-digestion patterns in figure 3. Without such a control, it is difficult to evaluate the robustness of the data. The T1-cleavage patterns are indeed different between the absence and presence of Hdp1, but the results are messy and inconclusive, as the authors note themselves. This is critical because the majority of the paper's novelty comes from the data in this figure.

*Reviewer #3 (Recommendations for the authors):*

A very clearly written and enjoyable story with clean experiments. Several suggestions to address the weaknesses pointed out in the public review:

1. None of the growth data were shown although it was extensively discussed in the results text. I don't think you need all of it (especially all the random mutagenesis assays), but at least including growth data (e.g. growth curves in liquid minimal media or images of spot assays on minimal media plates) for the Hdp1 L27Q-expressing strain compared to the Hdp1-expressing strain and the and empty vector strain would make it easier for the reader to see that it is suitable as a non-functional control sequence.

2. A control experiment that expands Sup. Figure 2B to include Hdp1opt L27Q's effects on lacZ reporter expression would support its use as a non-functional variant in the EMSA experiments. Hdp1 and Hdp1opt seem very different in sequence, hydrophobicity, and function. Although EMSA shows that Hdp1opt L27Q binds the his operator with lower affinity than Hdp1opt, data indicating that this result is consistent with Hdp1opt L27Q's behavior in vivo would strengthen the paper.

3. Although the β-gal assays indicate that Hdp activation is likely specific to the his operon by comparing induction to the trp operon lacZ fusion, a control demonstrating this specificity in vitro would support that Hdp1 is not non-specifically binding mRNA. Using the thr operator sequence or another random section of similarly-sized RNA in an EMSA could demonstrate this.

---

## [Author Response]

Essential revisions:1. Add a table with a list of the Hdp1 and Hdp2 random mutants assayed, including any that retained function.

We have added a comprehensive list of the isolated Hdp1 and Hdp2 mutant variants, including information on the functionality, in revised Supplementary file 1b.

2. Soften the conclusions regarding the putative shared amino acid motif (Figure 1D).

We agree that the relevance of the discovered putative motif has not been conclusively established. We have softened our conclusions on this in the Results section (lines 163-174) as well as our Discussion on this (lines 485-490) to better emphasize the putative nature of the identified stretch of sequence similarity.

3. Include all relevant bacterial growth data.

All relevant bacterial growth data have been added to Figure 1G and Figure 1—figure supplement 1-3, 5.

4. Include a negative control RNA in the EMSA experiment.

We agree that this negative control is necessary to strengthen our claim on specificity. We have now included EMSA data with the *thr* operator RNA as a negative control (Figure 3B,C, Figure 3—figure supplement 3).

5. Add reporter assay data for Hdp1opt and Hdp1opt L27Q using one of the lacZ reporter fusions, to establish whether the L27Q mutation in the context of Hdp1opt disrupts function.

We have included this control in Figure 3—figure supplement 1, showing that the L27Q mutation in Hdp1_opt_ indeed causes a loss of functionality. We thank the reviewer for this important suggestion.

6. Add a replicate of the RNase T1 experiment; quantify the data.

We have included two additional RNase T1 probing experimental replicates in Supplementary Figure 3figure supplement 4, both of which demonstrate cleavage patterns similar to that of the experiment presented in the main text Figure 3D. Furthermore, we have clarified in the text that the main conclusion drawn from these experiments is that the *his* operator RNA undergoes a conformational change in the presence of Hdp1, and we have modified any additional statements regarding the probing data to account for the limitations of the experiment (lines 419-431). Because we only demonstrate and conclude that a general conformational change occurs upon protein binding and the three experimental replicates all present similar results, we do not think quantifying the probing data would provide any additional information.

7. Demonstrate a specific interaction of Hdp1 (or Hdp1opt) with the his mRNA 5' UTR in vivo. If this involves a tagged Hdp1, include controls to show that the tagged protein retains activity in vivo.

This is an excellent suggestion and we previously had difficulty adding protein tags to Hdp1 without compromising its rescue activity. Fortunately, we were successful in generating an HA-tagged variant that retained the ability to rescue SerB loss. Using this construct, we performed in vivo pull-down assays with tagged versions of Hdp1 and Hdp1 L27Q. Excitingly, our co-IP results are in agreement with the data from our proteomics, EMSA, and Β-galactosidase activity experiments and demonstrate that (i) Hdp1 interacts with RNA transcripts regulated by the *his* operator in vivo, (ii) Hdp1 exhibits reduced binding to the control *thr* operator RNA, and (iii) the Hdp1 L27Q mutant also shows reduced interactions with the *his* operator RNA. These data are reported in Figure 3A and described in lines 313-337. We thank the reviewers for this suggestion, as it has substantially strengthened our conclusions.

Please also respond to comments made by individual reviewers.Reviewer #1 (Recommendations for the authors):1. The authors describe several results involving the growth of different strains, e.g. growth rescue of the serB deletion by the three small proteins, and growth of hisB/serA/serC mutants. These data should be shown.

We have added all relevant growth data in Figure 1G and Figure 1—figure supplement 1-3, 5.

2. The authors describe random mutagenesis of Hdp1 and Hdp2. A full list of the identified mutations should be included as a table. Were any mutants identified and sequenced that maintained function? If so, these should be included in the table.

We have included a list of all identified mutations and their effect on functionality in Supplementary file 1b.

3. The motif shown in Figure 1D may be important, but the evidence supporting this idea is relatively weak. The sequences are somewhat divergent between the three proteins, and the previously reported protein SynSerB3 lacks a match to this motif. Conclusions about the importance of the motif should be softened.

We fully agree on the putative nature of the identified region of sequence similarity and have adapted phrasing and interpretation to more accurately represent this in both the Results (lines 163-174) and Discussion (lines 485-490).

4. The Hdp1opt L27Q mutant should be tested for its effect in vivo with the reporter assay.

We have now included this important control and show that the Hdp1opt L27Q mutant does not increase Β-galactosidase activity in the in vivo reporter assay (Figure 3—figure supplement 1). Furthermore, we now show that *his* operator-regulated RNA transcripts are less enriched in pull-down experiments performed with the HA-tagged Hdp1 L27Q mutant variant (Figure 3A). We thank the reviewer for this important suggestion, as it strengthens our manuscript’s conclusions.

Reviewer #2 (Recommendations for the authors):– Currently, all growth/fitness data are not presented. Growth curves or spot plate assays for all relevant experiments should be added, among which for Hpd1-3 random proteins, Hpd1-3 versions with premature stop codons or truncations, recoded Hdp ORFs, Hdp1(L27Q), Hdp1-optimized, Hdp1-optimized(L27Q), claim that cells expressing Hdp2 grow slower than cells expressing Hdp1 or Hdp3, experiments in ΔserA, ΔserC, ΔYtjC, Δgph, or ΔhisB backgrounds.These experiments should include the following controls: (i) an empty vector, (ii) a SerB complementation on the same vector, and (iii) a HisB complementation on the same vector.The fitness of cells expressing Hpd1-3 in medium supplemented with serine should also be included to show their cost/toxicity.

We have added the requested growth data (Figure 1G, Figure 1—figure supplement 1-3, 5).

– The authors are encouraged to clearly state the model for Hdps molecular function before describing the data in figure 2 to help the readers evaluate their findings. If I understand correctly, the model suggests that when histidine-charges tRNA levels are high, there are low transcript levels of the his operon. Expression of Hdps results in increased RNA levels of the his operon, leading to increased protein levels of HisB that compensate for the SerB deletion. To test this model the authors should:– Demonstrate that in a medium without histidine and serine, HisB protein levels are high enough to allow the growth of ΔserB cells.– Demonstrate that in a medium without histidine both the fitness improvement and the increase in HisB protein levels as a result of Hdps expression are dramatically reduced because HisB levels are already high due to activation of the his operon and the effects of Hdps are partially/completely alleviated.

It has been previously established (for examples see Patrick, et al. PMID: 17884825, Digianantonio and Hecht PMID: 26884172) that overexpression of HisB restores growth of a Δ*serB* mutant on minimal medium. We have now included this control experiment as well, showing overexpression of HisB from pRD2 allows growth (Figure 1—figure supplement 2, 3, 5). It is also known that a Δ*serB* mutant is auxotrophic, and does not grow on minimal medium lacking both serine and histidine. If lack of histidine alone would allow growth restoration, the mutant would not classify as auxotrophic on M9 minimal medium, which does not contain serine and histidine. The inability of the *serB* mutant carrying an empty control plasmid to grow on M9 is now shown in Figure 1G. We established that HisB is upregulated upon expression of the Hdps and we have established that HisB is required for growth restoration of a Δ*serB* mutant by the Hdps. We believe our current work and data from previous studies clearly show that the mechanistic basis for which Δ*serB* auxotrophy is rescued by HisB upregulation is not achieved by histidine starvation alone.

– It is unclear if LacZ levels are statistically significant between an empty vector and Hdp2 expression. The authors should clarify this issue and also directly measure HisB protein levels upon Hdp2 expression. This is critical to support the claim that Hdp2 functions similarly to Hdp1 and Hdp3.

We have now included total proteome analyses for Hdp1, Hdp2, Hdp3, and the Hdp1 L27Q mutant (Figure 2C, Figure 2—figure supplement 2). While Hdp1 and Hdp3 show a specific and significant increase in *his* protein abundance, we were not able to detect a similar increase upon overexpression of Hdp2 in rich medium. This is in agreement with the Β-galactosidase activity assays with the *lacZ* reporter fusion, where we only measure a modest, yet statistically significant, increase in Β-galactosidase activity with Hdp2. Notably, the Δ*serB* mutant expressing Hdp2 also exhibits the weakest rescue phenotype/slowest growth on minimal medium. We have included this growth data in Figure 1G and Figure 1—figure supplement 2, 3, 5. While it is possible that Hdp2 acts via a completely different mechanism, we believe it is highly unlikely, given Hdp1 and Hdp3, as well as the SynserB3 isolated by Digianantonio and Hecht, all likely act via the mechanism presented in this manuscript. As mentioned above, it is not possible to determine HisB levels under non-permissive conditions because the empty plasmid control does not permit growth (see response to public review). Given its similar characteristics to Hdp1 and Hdp3, it is our opinion that Hdp2 also functions by the same mechanism. However, the upregulation of HisB may be less pronounced, and possibly only significant under non-permissive rescue conditions. Because of its strong rescue phenotype, we have focused our study on Hdp1 as a representative de novo gene rescuing a Δ*serB* auxotroph, and we do not believe it is necessary to prove the mechanistic basis of rescue for all 3 variants. The possibility that Hdp2 or Hdp3, albeit very unlikely, act via different pathways does not affect the main objective of our study, namely that we identified a de novo-generated protein that rescues Δ*serB* auxotrophy by upregulating expression via direct RNA-binding interactions. Nevertheless, we do acknowledge that it is theoretically possible that Hdp2 functions via a different mechanism and we have now mention this in the Results (lines 279-280).

– LacZ levels for cells expressing Hdp1(L27Q) seem very similar to LacZ levels for cells expressing Hdp2, however, the authors claim that the former is non-functional and the latter is functional. The authors should show growth data, provide statistical analysis for this experiment, and directly measure HisB protein levels for each construct. This is critical for the claim that increasing HisB levels is the mechanism of rescue by Hdps.

We have now added growth data for all relevant constructs including Hdp2 (Figure 1G, Figure 1—figure supplement 1-3, 5), and conducted proteomics analyses for for Hdp1, Hdp2, Hdp3, and the Hdp1 L27Q mutant, including a statistical analysis (Figure 2C, Figure 2—figure supplement 2). As discussed above, we are confident that we have conclusively demonstrated that Hdp1 rescues SerB loss by upregulation of HisB. Thus, the suggested measurements of HisB protein levels for each construct will not add anything with regard to understanding how the proteins act mechanistically. We have amended the text to state an alternative mechanism is theoretically possible, even though highly unlikely, in lines 279-280.

– Figure 2E does not support the authors' claim because constructs (ii)-(iv) show high basal expression levels. Therefore, any Hdp1 effect might be negligible in this context. The authors should find mutations in the his operator that maintain its functionality on one hand but alleviate Hdp1's function. Because this claim is central to the model presented in the paper, this control is very important.

It is an interesting and important point that the reviewer raises and it is one that we independently considered. First, it should be noted that the data in Figure 2E, to which the reviewer refers, was only intended to show that the ability of Hdp1 to upregulate expression requires the intact 5’ regulatory region (described in lines 296-302). Thus, it was not it intended to address the exact mechanism with regard to where Hdp1 binds and how it influences the *his* operator RNA. Second, it is not straightforward to perform the actual experiments that the reviewer suggests and identify a specific binding site, since it assumes that such mutations exist (which they might not if, for example, the binding site for Hdp1 involves the terminator structure). Second, if they do exist, they are probably very rare since a majority of the mutations in the operator region are expected to affect functionality (by influencing the different secondary structures). Third, even if we find such mutations, they would not necessarily identify the exact binding site, since the Hdp1 binding site could involve a secondary/tertiary structure and a point mutation could directly/indirectly affect this. Thus, proper execution of this experiment would require extensive in vivo probing of secondary/tertiary mRNA structures of operator mutants (of the type requested by the reviewer, again, assuming they exist) in the absence and presence of Hdp1. In conclusion, even though the idea is valid, we think the experiments required are very extensive and far beyond the objectives of the current manuscript.

– The thr-lacZ construct is not a good control for the experiment in figure 2B because its basal LacZ levels are high in the conditions of the experiment (similar to my previous comment). Yet, this is a critical control, so the authors should find a different operator with a similar structure and function to the his operator and for which LacZ basal levels are low for an empty vector control. If LacZ levels remain low upon Hdp1 expression, that will be an important support for the authors' model.

The *thr* operator region negatively regulates expression of *thr* operon structural genes by terminator/antiterminator formation in a manner similar to that of the *his* operator, which therefore represents a suitable control to assess the specificity of the Hdp-dependent *his* operon upregulation. In rich medium, transcription of the threonine operon is repressed. As expected, baseline transcription might vary between different operators and their subsequent operons. However, the fact that the *thr* operator has a higher baseline expression in a repressed state does not have any bearing on the conclusion that Hdp1 does in fact not de-repress expression of the *thr* operon. While we consider this finding to be conclusive by itself, it is further confirmed by the specific and significant upregulation of the *his* operon shown in the whole proteome analysis, where any off target de-repression would be detected (e.g. *thr*-encoded proteins). The newly added *thr* operator RNA control in the EMSAs (Figure 3B,C and Figure 3—figure supplement 3), as well as the pull-down assay (Figure 3A) further confirm the *his*-specific action of Hdp1.

– Such an alternative operator control should be used as the control for the EMSA experiments and T1-digestion patterns in figure 3. Without such a control, it is difficult to evaluate the robustness of the data. The T1-cleavage patterns are indeed different between the absence and presence of Hdp1, but the results are messy and inconclusive, as the authors note themselves. This is critical because the majority of the paper's novelty comes from the data in this figure.

We thank the reviewer for the suggestion of adding this important control. We have now included EMSA experiments with the *thr* operator RNA as a negative control and confirm that Hdp1 demonstrates an almost 3-fold reduction in binding affinity to this RNA, in comparison to that with the *his* operator RNA. These data are now presented in Figure 3B,C and Figure 3—figure supplement 3. The *thr* operator RNA is also included as a control in the Β-galactosidase activity assays, proteomics, and co-IP experiments.

We agree that the RNase T1-digestion alone does not provide conclusive evidence, and we have modified the interpretation and discussion of the data accordingly and we now more clearly address the limitations of the experiment in the text (lines 419-431). We have also added two additional T1 probing replicates (Figure 3—figure supplement 4), which are in line with the results presented in the main text figure.

Reviewer #3 (Recommendations for the authors):A very clearly written and enjoyable story with clean experiments. Several suggestions to address the weaknesses pointed out in the public review:1. None of the growth data were shown although it was extensively discussed in the results text. I don't think you need all of it (especially all the random mutagenesis assays), but at least including growth data (e.g. growth curves in liquid minimal media or images of spot assays on minimal media plates) for the Hdp1 L27Q-expressing strain compared to the Hdp1-expressing strain and the and empty vector strain would make it easier for the reader to see that it is suitable as a non-functional control sequence.

This is a valid point. We have fixed this and added all relevant bacterial growth data in Figure 1G and Supplementary Figure 1—figure supplement 1-3, 5.

2. A control experiment that expands Sup. Figure 2B to include Hdp1opt L27Q's effects on lacZ reporter expression would support its use as a non-functional variant in the EMSA experiments. Hdp1 and Hdp1opt seem very different in sequence, hydrophobicity, and function. Although EMSA shows that Hdp1opt L27Q binds the his operator with lower affinity than Hdp1opt, data indicating that this result is consistent with Hdp1opt L27Q's behavior in vivo would strengthen the paper.

The reviewer raises a valid concern. We have now included the necessary control in Figure 3—figure supplement 1 and show consistent phenotypes for Hdp1 L27Q. Furthermore, we have also performed total proteome analysis for Hdp1 L27Q (Figure 2C, Figure 2—figure supplement 2), as well as a co-IP experiment with the Hdp1 L27Q mutant, which also demonstrates a clear difference in *his* operator transcript binding capacity in vivo (Figure 3A).

3. Although the β-gal assays indicate that Hdp activation is likely specific to the his operon by comparing induction to the trp operon lacZ fusion, a control demonstrating this specificity in vitro would support that Hdp1 is not non-specifically binding mRNA. Using the thr operator sequence or another random section of similarly-sized RNA in an EMSA could demonstrate this.

We appreciate the reviewer’s thorough suggestions. We have added this control and confirmed reduced binding affinity for the *thr* operator RNA (Figure 3B,C, Figure 3—figure supplement 3). The inclusion of this control as suggested by all reviewers has significantly strengthened the central claim of our study.